# Don't Play Favorites:
# Minority Guidance for Diffusion Models

**Soobin Um, Suhyeon Lee & Jong Chul Ye**
KAIST, Daejeon, Republic of Korea
{sum,suhyeon.lee,jong.ye}@kaist.ac.kr

## Abstract

We explore the problem of generating minority samples using diffusion models. The minority samples are instances that lie on low-density regions of a data manifold. Generating a sufficient number of such minority instances is important, since they often contain some unique attributes of the data. However, the conventional generation process of the diffusion models mostly yields majority samples (that lie on high-density regions of the manifold) due to their high likelihoods, making themselves ineffective and time-consuming for the minority generating task. In this work, we present a novel framework that can make the generation process of the diffusion models focus on the minority samples. We first highlight that Tweedie's denoising formula yields favorable results for majority samples. The observation motivates us to introduce a metric that describes the uniqueness of a given sample. To address the inherent preference of the diffusion models w.r.t. the majority samples, we further develop *minority guidance*, a sampling technique that can guide the generation process toward regions with desired likelihood levels. Experiments on benchmark real datasets demonstrate that our minority guidance can greatly improve the capability of generating high-quality minority samples over existing generative samplers. We showcase that the performance benefit of our framework persists even in demanding real-world scenarios such as medical imaging, further underscoring the practical significance of our work. Code is available at https://github.com/soobin-um/minority-guidance.

## 1 Introduction

Conventional large-scale datasets typically exhibit long-tailed distributions, where the majority of samples are concentrated in high-probability regions of a data manifold (Ryu et al., 2017; Liu et al., 2019). While in low-likelihood regions, there are instances called *minority* samples, which contain unique attributes rarely observed in the majority of the data. The ability to generate minority data holds substantial relevance in critical real-world applications, particularly in enhancing the predictive capabilities of medical diagnosis models by introducing additional instances of rare conditions. This augmentation is also significant in promoting fairness, aligning with social vulnerabilities often associated with minority instances. Furthermore, the unique features contained within low-likelihood instances are pivotal in applications like creative AI (Han et al., 2022), where the generation of samples with exceptional creativity is an essential component of the task.

One challenge is that generation focused on such minority samples is actually difficult to perform (Hendrycks et al., 2021). This holds even for diffusion-based generative models (Sohl-Dickstein et al., 2015; Ho et al., 2020) that provide strong coverage for a given data distribution (Sehwag et al., 2022). The generation process of the diffusion models can be understood as simulating the reverse of a diffusion process that defines a set of noise-perturbed data distributions (Song & Ermon, 2019). The principle guarantees their generated samples to respect the (clean) data distribution, which naturally makes the sampler become *majority-oriented*, i.e., producing higher likelihood samples more frequently than lower likelihood ones (Sehwag et al., 2022), when deployed on the long-tail-distributed data. Consequently, collecting even a handful number of minority samples using diffusion models often requires meticulous curation and substantial investments in terms of time and computational resources (Sehwag et al., 2022).

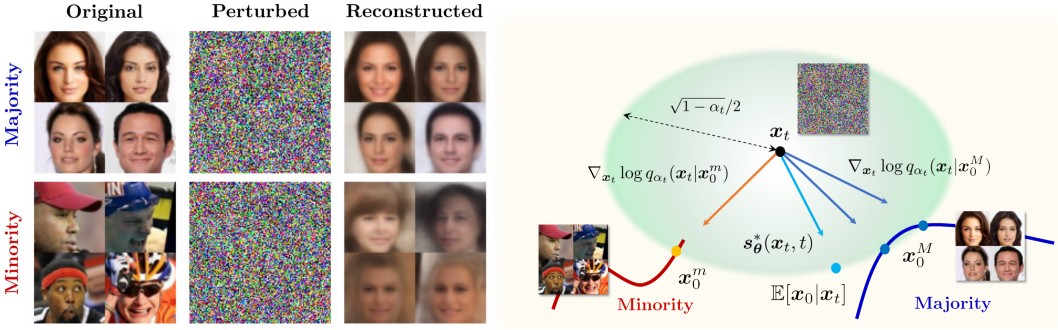

(a) Biased denoising on CelebA (Liu et al., 2015)  (b) Geometric intuition

Figure 1: Inherent bias of Tweedie-based denoiser toward majority features. **(a)** (Left column) Clean images $x_0$ from CelebA; (Middle column) Noised samples $x_t$ made by the DDPM perturbation (i.e., Eq. (1)) with $t = 0.9T$ on the clean versions in the left column; (Right column) Denoised samples $\hat{x}_0$ using Tweedie's formula (i.e., Eq. (6)) on the perturbed ones. The top (bottom) row represents the perturbation-denoising sequence performed on majority (minority) featured samples. **(b)** Geometric interpretation of the bias based on optimal score expression $s_{\boldsymbol{\theta}}^*(x_t, t) = \mathbb{E}_{q_{\alpha_t}(x_0|x_t)}[\nabla_{x_t} \log q_{\alpha_t}(x_t|x_0)]$.

**Contribution.** In this work, we propose a novel framework to counteract the majority-focused nature of diffusion models. Our approach begins with the development of a new metric to describe the uniqueness of features in a given sample. Our metric, which we call *minority score*, is based on Tweedie's formula (Robbins, 1992) that yields the posterior mean of a clean sample given a noise-corrupted one (Kim & Ye, 2021; Chung et al., 2022a). We emphasize that given a pretrained diffusion model trained on long-tailed data, Tweedie's formula leads to biased denoising against minority samples. Specifically for strong noise-perturbation, it often incurs significant information loss for low-likelihood minority samples (see Figure 1a for instance). This motivates us to define our metric as a discrepancy between clean samples and its reconstructed versions via Tweedie's formula. We highlight that the proposed metric is efficient to compute, requiring fewer network evaluations than other diffusion-based outlier metrics (Wolleb et al., 2022; Wyatt et al., 2022; Teng et al., 2022). We also establish a connection to the noise-matching error of diffusion models, offering an additional insight on the effectiveness of our metric for detecting low-density instances.

Given the proposed metric at hand, we develop a sampling technique that can be used for encouraging diffusion models to produce minority-featured samples. Our sampler, which we call *minority guidance*, conditions the sampling process on a desired level of minority score using the classifier guidance technique (Dhariwal & Nichol, 2021). One key advantage of minority guidance lies in its label-agnostic nature, which is contrary to existing minority samplers that rely upon the use of labeled data (Yu et al., 2020; Lin et al., 2022; Sehwag et al., 2022).

To validate the efficacy of our proposals, we conduct comprehensive experiments across a diverse range of real-world benchmarks, which encompasses both unconditional and conditional datasets. Specifically to highlight the practical importance of our work, we also explore the domain of medical imaging where generating minority instances is a demanding issue in the area (e.g., synthesis of highly-distinctive lesion images). To that end, we first demonstrate that minority score is effective for identifying low-density instances, sharing similar trends with existing criteria like Average k-Nearest Neighbor (AvgkNN) and Local Outlier Factor (LOF) (Breunig et al., 2000). We also exhibit that our minority guidance can serve as an effective knob for controlling the uniqueness of features for generated samples. We further demonstrate that our sampler greatly improves the ability to generate high-quality minority samples on the considered real benchmarks including the medical imaging dataset, as demonstrated by high values of outlier measures (such as AvgkNN and LOF) and better quality metrics like Fréchet Inception Distance (Heusel et al., 2017).

**Related work.** Generating minority samples that contain novel features has been explored under a number of different scenarios (Sehwag et al., 2022; Lee et al., 2022; Yu et al., 2020; Lin et al., 2022; Qin et al., 2023). The closest instance to our work is Sehwag et al. (2022) wherein the authors propose a sampling technique for diffusion models, which can encourage the generation process to move toward low-density regions w.r.t. a specific class using a class-predictor and a conditional

model. The key distinction w.r.t. ours is that their method is limited to class-conditional settings and therefore does not work with unlabeled data. Another notable work that bears an intimate connection to ours is Lee et al. (2022). As in Sehwag et al. (2022), they also leverage diffusion models yet for a different modality, graph data (e.g., lying in chemical space). The key idea therein is to produce out-of-distribution (OOD) samples via a customized generation process designed for maintaining some desirable properties w.r.t. the focused data space (e.g., plausible chemical structures). Since it is tailored for a particular modality which is inherently distinct from our focus (e.g., image data), hence not directly comparable to our approach. The authors in (Yu et al., 2020; Lin et al., 2022; Qin et al., 2023; Huang & Jafari, 2023) focus on improving data coverage for low-density instances via the use of labels that indicate minority instances. A distinction w.r.t. ours is that their methods require the knowledge on the minority labels. Samuel et al. (2023) develops a sampler for text-to-image (T2I) diffusion models (Rombach et al., 2022) to specifically enhance the quality of generated samples prompted with unique concepts (e.g., shaking hands). However, their method is confined to text-paired datasets and not applicable to more general benchmarks like unconditional data.

Leveraging diffusion models for identifying uncommon instances (e.g., OOD samples) has recently been proposed, especially in the context of medical imaging (Wolleb et al., 2022; Wyatt et al., 2022; Teng et al., 2022). Their methods share a similar spirit as ours: measuring discrepancies between the original and reconstructed images. However, their focus is for the anomaly detection for a given data by generating the majority (normal) samples, which is different from our focus.

## 2 BACKGROUND

Before delving into details of our work, we briefly review several key concepts related to diffusion-based generative models. We provide an overview of diffusion models with a particular focus on Denoising Diffusion Probabilistic Models (DDPM) (Ho et al., 2020) that our framework is based upon. Also, we cover Tweedie's denoising formula (Robbins, 1992) that plays an important role in developing our metric. We review the classifier guidance (Dhariwal & Nichol, 2021) which provides a basic principle for our sampling technique.

### 2.1 DIFFUSION-BASED GENERATIVE MODELS

Diffusion-based generative models (Sohl-Dickstein et al., 2015; Song & Ermon, 2019) are latent variable models defined by a forward diffusion process and the associated reverse process. The forward process is a Markov chain with a Gaussian transition where data is gradually corrupted by Gaussian noise in accordance with a (positive) variance schedule $\{\beta_t\}_{t=1}^T$: $q(\boldsymbol{x}_t \mid \boldsymbol{x}_{t-1}) :=$ $\mathcal{N}(\boldsymbol{x}_t; \sqrt{1 - \beta_t}\boldsymbol{x}_{t-1}, \beta_t\boldsymbol{I})$ where $\{\boldsymbol{x}_t\}_{t=1}^T$ are latent variables having the same dimensionality as data $\boldsymbol{x}_0 \sim q(\boldsymbol{x}_0)$. One notable property is that the forward process enables *one-shot* sampling of $\boldsymbol{x}_t$ at any desired timestep $t$:

$$q_{\alpha_t}(\boldsymbol{x}_t \mid \boldsymbol{x}_0) = \mathcal{N}(\boldsymbol{x}_t; \sqrt{\alpha_t}\boldsymbol{x}_0, (1 - \alpha_t)\boldsymbol{I}), \tag{1}$$

where $\alpha_t := \prod_{s=1}^t (1 - \beta_s)$. The variance schedule is designed to respect $\alpha_T \approx 0$ so that $\boldsymbol{x}_T$ becomes approximately distributed as $\mathcal{N}(\boldsymbol{0}, \boldsymbol{I})$. The reverse process is another Markov Chain that is parameterized by a *learnable* Gaussian transition: $p_{\boldsymbol{\theta}}(\boldsymbol{x}_{t-1} \mid \boldsymbol{x}_t) := \mathcal{N}(\boldsymbol{x}_{t-1}; \boldsymbol{\mu_\theta}(\boldsymbol{x}_t, t), \beta_t\boldsymbol{I})$. One way to express $\boldsymbol{\mu_\theta}(\boldsymbol{x}_t, t)$ is to employ a noise-conditioned score network $\boldsymbol{s_\theta}(\boldsymbol{x}_t, t) :=$ $\nabla_{\boldsymbol{x}_t} \log p_{\boldsymbol{\theta}}(\boldsymbol{x}_t)$ that approximates the score function $\nabla_{\boldsymbol{x}_t} \log q_{\alpha_t}(\boldsymbol{x}_t)$. Then, we have $\boldsymbol{\mu_\theta}(\boldsymbol{x}_t, t) = \frac{1}{\sqrt{1-\beta_t}}(\boldsymbol{x}_t + \beta_t\boldsymbol{s_\theta}(\boldsymbol{x}_t, t))$ (Song & Ermon, 2019; Song et al., 2020b). The score network is trained with a weighted sum of denoising score matching (DSM) (Vincent, 2011) objectives:

$$\min_{\boldsymbol{\theta}} \sum_{t=1}^T w_t \mathbb{E}_{q(\boldsymbol{x}_0)q_{\alpha_t}(\boldsymbol{x}_t|\boldsymbol{x}_0)}[\|\boldsymbol{s_\theta}(\boldsymbol{x}_t, t) - \nabla_{\boldsymbol{x}_t} \log q_{\alpha_t}(\boldsymbol{x}_t \mid \boldsymbol{x}_0)\|_2^2], \tag{2}$$

where $w_t := 1 - \alpha_t$. Notably, this procedure is equivalent to building a noise-prediction network $\boldsymbol{\epsilon_\theta}(\boldsymbol{x}_t, t)$ that regresses noise added on clean data $\boldsymbol{x}_0$ through the forward process in Eq. (1) (Vincent, 2011; Song et al., 2020b). This establishes an intimate connection between the two networks: $\boldsymbol{s_\theta}(\boldsymbol{x}_t, t) = -\boldsymbol{\epsilon_\theta}(\boldsymbol{x}_t, t)/\sqrt{1 - \alpha_t}$, implying that the score model is closely related to a denoiser. Once obtaining the optimal model via the DSM training, data generation can be done by starting

from $\boldsymbol{x}_T \sim \mathcal{N}(\boldsymbol{0}, \boldsymbol{I})$ and following the reverse Markov Chain down to $\boldsymbol{x}_0$:

$$\boldsymbol{x}_{t-1} = \frac{1}{\sqrt{1-\beta_t}} (\boldsymbol{x}_t + \beta_t \boldsymbol{s}_{\boldsymbol{\theta}}^*(\boldsymbol{x}_t, t)) + \beta_t \boldsymbol{z}, \quad \boldsymbol{z} \sim \mathcal{N}(\boldsymbol{0}, \boldsymbol{I}), \tag{3}$$

which is often called *ancestral sampling* (Song et al., 2020b). This process corresponds to a discretized simulation of a stochastic differential equation that defines $\{p_{\boldsymbol{\theta}}(\boldsymbol{x}_t)\}_{t=0}^T$ (Song et al., 2020b), which guarantees to sample from $p_{\boldsymbol{\theta}}(\boldsymbol{x}_0) \approx q(\boldsymbol{x}_0)$.

## 2.2 TWEEDIE'S FORMULA FOR DENOISING

Given a perturbed sample with exponential family noise, a classic result of Tweedie's formula gives the Bayes optimal solution for denoising (i.e., the posterior mean) using the score function (Robbins, 1992; Kim & Ye, 2021). Under Gaussian perturbation $q_\sigma(\tilde{\boldsymbol{x}}|\boldsymbol{x}) = \mathcal{N}(\tilde{\boldsymbol{x}}; \boldsymbol{x}, \sigma^2 \boldsymbol{I})$, the formula reads: $\mathbb{E}[\boldsymbol{x} \mid \tilde{\boldsymbol{x}}] = \tilde{\boldsymbol{x}} + \sigma^2 \nabla_{\tilde{\boldsymbol{x}}} \log q_\sigma(\tilde{\boldsymbol{x}})$ where $q_\sigma(\tilde{\boldsymbol{x}}) := \int q_\sigma(\tilde{\boldsymbol{x}}|\boldsymbol{x}) q(\boldsymbol{x}) \, d\boldsymbol{x}$. Tweedie's formula can be extended to our focused DDPM perturbation (i.e., Eq. (1)) (Chung et al., 2022a;b):

$$\mathbb{E}[\boldsymbol{x}_0 \mid \boldsymbol{x}_t] = \frac{1}{\sqrt{\alpha_t}} \left( \boldsymbol{x}_t + (1 - \alpha_t) \nabla_{\boldsymbol{x}_t} \log q_{\alpha_t}(\boldsymbol{x}_t) \right). \tag{4}$$

Notice that once having the optimal score model $\boldsymbol{s}_{\boldsymbol{\theta}}^*(\boldsymbol{x}_t, t) = \nabla_{\boldsymbol{x}_t} \log q_{\alpha_t}(\boldsymbol{x}_t)$, we can implement the formula and get the posterior mean by using the optimal model in place of the score function in Eq. (4).

## 2.3 CLASSIFIER GUIDANCE FOR DIFFUSION MODELS

Suppose we have access to auxiliary classifier $p_{\boldsymbol{\phi}}(y|\boldsymbol{x}_t)$ that predicts class $y$ given perturbed input $\boldsymbol{x}_t$. The main idea of the classifier guidance is to construct the score function of a conditional density w.r.t. $y$ by mixing the score model $\boldsymbol{s}_{\boldsymbol{\theta}}(\boldsymbol{x}_t, t)$ and the log-gradient of the auxiliary classifier:

$$\begin{aligned} \nabla_{\boldsymbol{x}_t} \log \tilde{p}_{\boldsymbol{\theta}}(\boldsymbol{x}_t \mid y) &= \nabla_{\boldsymbol{x}_t} \{ \log p_{\boldsymbol{\theta}}(\boldsymbol{x}_t) + \log p_{\boldsymbol{\phi}}(y|\boldsymbol{x}_t)^w \} \\ &= \boldsymbol{s}_{\boldsymbol{\theta}}(\boldsymbol{x}_t, t) + w \nabla_{\boldsymbol{x}_t} \log p_{\boldsymbol{\phi}}(y|\boldsymbol{x}_t) \\ &=: \tilde{\boldsymbol{s}}_{\boldsymbol{\theta}}(\boldsymbol{x}_t, t, y), \end{aligned} \tag{5}$$

where $w$ is a hyperparameter that controls the strength of the classifier guidance. Employing the mixed score $\tilde{\boldsymbol{s}}_{\boldsymbol{\theta}}(\boldsymbol{x}_t, t, y)$ in place of $\boldsymbol{s}_{\boldsymbol{\theta}}(\boldsymbol{x}_t, t)$ in the generation process (e.g., in Eq. (3)) enables the conditional sampling w.r.t. $\tilde{p}_{\boldsymbol{\theta}}(\boldsymbol{x}_t|y) \propto p_{\boldsymbol{\theta}}(\boldsymbol{x}_t) p_{\boldsymbol{\phi}}(y|\boldsymbol{x}_t)^w$. Increasing the scaling factor $w$ affects the curvature of $p_{\boldsymbol{\phi}}(y|\boldsymbol{x}_t)^w$ around given $y$ to be more sharp, i.e., gives more strong focus on some noticeable features w.r.t. $y$, which often yields improvement of fidelity w.r.t. the corresponding class at the expense of diversity (Dhariwal & Nichol, 2021).

## 3 METHOD

We present our framework herein that specifically focuses on generating minority samples lying on low-density regions of a data manifold. To this end, we start by pinpointing the inherent bias of Tweedie's denoising formula against low-density samples. In light of this, we come up with a measure for describing the uniqueness of features. We subsequently develop a sampler that can guide the generation process of diffusion models toward minority regions. Throughout the section, we follow the setup and notations presented in Section 2.

## 3.1 INHERENT BIAS OF TWEEDIE-BASED DENOISER

Consider data distribution $q(\boldsymbol{x}_0)$ where unique features of data are contained in low-density regions[1]. Suppose we have access to diffusion model $\boldsymbol{s}_{\boldsymbol{\theta}}^*(\boldsymbol{x}_t, t)$ trained on dataset $\boldsymbol{x}_0 \sim q(\boldsymbol{x}_0)$ via DSM (in Eq. (2)). Then for Gaussian-perturbed sample $\boldsymbol{x}_t \sim q_{\alpha_t}(\boldsymbol{x}_t \mid \boldsymbol{x}_0)$, one can obtain the posterior mean $\hat{\boldsymbol{x}}_0$ by implementing Tweedie's formula in Eq. (4):

$$\hat{\boldsymbol{x}}_0 := \mathbb{E}[\boldsymbol{x}_0 \mid \boldsymbol{x}_t] = \frac{1}{\sqrt{\alpha_t}} \left( \boldsymbol{x}_t + (1 - \alpha_t) \boldsymbol{s}_{\boldsymbol{\theta}}^*(\boldsymbol{x}_t, t) \right). \tag{6}$$

---

[1]We note that such long-tailed property is common in conventional large-scale benchmarks (Ryu et al., 2017; Liu et al., 2019; Sehwag et al., 2022).

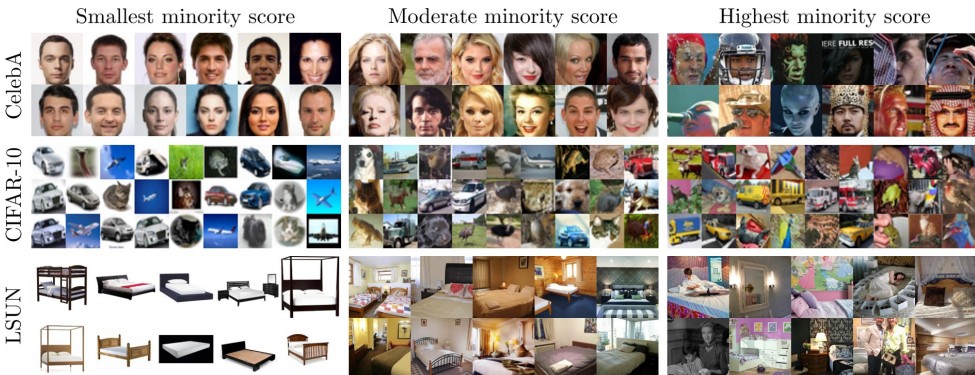

Figure 2: Real samples with the smallest (left column), moderate (middle column), and the highest (right column) minority score, our proposed metric for identifying minorities.

While it is the best effort in a statistical sense, this denoising has a critical downside in terms of fairness (Choi et al., 2020; Xiao et al., 2021). The issue is that the expectation induced by the formula makes the denoising process strongly affected by majority samples (that mostly contribute to the average), thereby yielding *biased* results against low-likelihood minority instances. The discrimination is often severe under strong perturbation (e.g., $t \approx T$) where denoised outputs become close to dataset mean (Karras et al., 2022).

Figure 1a illustrates an example of this observation for groups of majority and minority samples in CelebA (Liu et al., 2015). As we can see, the denoiser based on Tweedie's formula successfully restores some key structural properties (e.g., pose and gender) when perturbation is made upon majority samples. However, it fails to do so for minority-featured samples. Instead, the Tweedie-based denoiser embeds some common features in place of the unique attributes that are originally contained in the clean minority samples, exhibiting its inherent bias toward majority features. Figure 1b gives geometric illustration of the phenomenon. In the figure, we employ another expression for the optimal score function provided in the following proposition:

**Proposition 1.** *Consider the DSM optimization in Eq. (2). Assume that a given noise-conditioned score network $s_{\boldsymbol{\theta}}(\boldsymbol{x}_t, t)$ have enough capacity. Then for each timestep $t$, the optimality of the score network is achievable at:*

$$\boldsymbol{s}_{\boldsymbol{\theta}}^*(\boldsymbol{x}_t, t) = \mathbb{E}_{q_{\alpha_t}(\boldsymbol{x}_0|\boldsymbol{x}_t)} \left[ \nabla_{\boldsymbol{x}_t} \log q_{\alpha_t}(\boldsymbol{x}_t \mid \boldsymbol{x}_0) \right].$$

See Section A.1 for the proof. Observe in Figure 1b that the optimal score function gives the averaged conditional score over the data distribution (conditioned on noised input $\boldsymbol{x}_t$), thereby producing directions being inclined to the manifold w.r.t. the majority samples.

## 3.2 MINORITY SCORE: MEASURING THE UNIQUENESS

Inspired by the discriminative treatment against minority features, we develop a metric that quantifies the uniqueness of features of a given data instance. Remember that Tweedie's formula often incurs significant loss of information for minority samples (as seen in Figure 1a). Hence, we employ a distance between original clean sample $\boldsymbol{x}_0$ and its denoised version $\hat{\boldsymbol{x}}_0$:

$$l(\boldsymbol{x}_0; \boldsymbol{s}_{\boldsymbol{\theta}}^*) := \mathbb{E}_{q_{\alpha_t}(\boldsymbol{x}_t|\boldsymbol{x}_0)}[d(\boldsymbol{x}_0, \hat{\boldsymbol{x}}_0)], \tag{7}$$

where $d(\cdot, \cdot)$ denotes a discrepancy measure (e.g., Learned Perceptual Image Patch Similarity (Zhang et al., 2018), squared-error loss), and $t$ is a timestep used for perturbation; see Sections C and D.1 for details on our choices of $d(\cdot, \cdot)$ and $t$. The expectation is introduced for randomness that comes from the DDPM perturbation[2]. We call this metric *minority score*, since it would yield high values for minorities that contain novel attributes.

Figure 2 visualizes the effectiveness of minority score on several benchmarks that have diverging characteristics. Notice that the samples in the left column exhibit features that are commonly observed in the corresponding datasets (e.g., frontal-view faces in CelebA (Liu et al., 2015)). On the

---

[2]We empirically found that the metric performs well even when the expectation is computed with a single sample from $q_{\alpha_t}(\boldsymbol{x}_t|\boldsymbol{x}_0)$. See Figure 2 for instance, where the computations are based on such single-sampling.

other hand, we see some unique features in the right column, such as "Eyeglasses" and "Wearing_Hat" that are famously known as minority attributes of CelebA (Amini et al., 2019; Yu et al., 2020). We found that minority score shares similar trends with existing outlier metrics, exhibiting its effectiveness for identifying minorities; see Section D.2 for empirical evidence. Thanks to the one-shot reconstruction offered by Tweedie's formula, minority score is more efficient to compute than previous methods that rely upon iterative forward and reverse diffusion processes so requiring a number of evaluations of models (Wolleb et al., 2022; Wyatt et al., 2022; Teng et al., 2022). In contrast, our metric requires only a few number of function evaluations. Given a conditional model and labeled data, minority score can be extended to class-conditional settings for identifying low-density samples w.r.t. a specific class; see Section B for details.

**Connection to noise-prediction error.** We found that minority score bears an intimate connection to the noise-estimation error of diffusion models. Specifically with the squared-error distance loss, minority score is equivalent (up to scaling) to the noise-matching error for timestep $t$. This offers additional insights on why low-density instances could yield high minority scores, e.g., since they would experience significant error in noise-estimation (compared to the majorities) due to their scarcity in data. Below we provide a formal statement of the claim. See Section A.2 for the proof.

**Corollary 1.** *Minority score, when defined with the squared-error distance loss $\|\cdot\|_2^2$, is equivalent to the noise prediction error w.r.t. timestep $t$ up to scaling:*

$$l(\boldsymbol{x}_0; \boldsymbol{s}_{\boldsymbol{\theta}}^*) = \tilde{w}_t \mathbb{E}_{p(\boldsymbol{\epsilon})} \left[ \|\boldsymbol{\epsilon}_{\theta}^*(\boldsymbol{x}_t, t) - \boldsymbol{\epsilon}\|_2^2 \right],$$

*where $\tilde{w}_t := (1 - \alpha_t)/\alpha_t$.*

### 3.3 MINORITY GUIDANCE: TACKLING THE PREFERENCE

Here a natural question arises. Given minority score at hand, what can we do for tackling the majority-focused generation of diffusion models so that they become more likely to generate the novel-featured samples? To address this question, we take a conditional generation approach[3] where we incorporate minority score as a conditioning variable into the generation process, which can then serve to produce the unique-featured samples by conditioning w.r.t. high minority score values. To condition the given framework with minimal effort, we employ the classifier guidance (Dhariwal & Nichol, 2021) that does not require re-building of class-conditional models. Below we describe in detail how we develop the conditional generation framework for minority score based on the classifier guidance technique.

Let us consider dataset $\boldsymbol{x}_0^{(1)}, \ldots, \boldsymbol{x}_0^{(N)} \overset{\text{i.i.d.}}{\sim} q(\boldsymbol{x}_0)$ and pretrained diffusion model $\boldsymbol{s}_{\boldsymbol{\theta}}^*(\boldsymbol{x}_t, t)$. For each sample, we compute minority score via Eq. (7) and obtain $l^{(1)}, \ldots, l^{(N)}$ where $l^{(i)} := l(\boldsymbol{x}_0^{(i)}; \boldsymbol{s}_{\boldsymbol{\theta}}^*), i \in \{1, \ldots, N\}$. We process the (positive-valued) minority scores as ordinal data with $L$ categories by thresholding them with $L-1$ levels of minority score. This yields the ordinary categorized minority scores $\tilde{l}^{(1)}, \ldots, \tilde{l}^{(N)} \in \{0, \ldots, L-1\}$ where $\tilde{l} = 0$ and $\tilde{l} = L - 1$ indicate the classes w.r.t. the most common and the rarest features, respec-

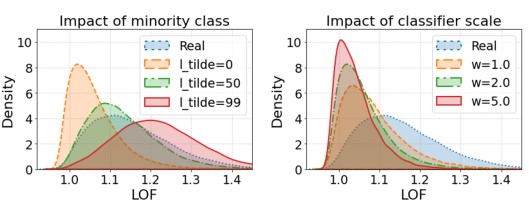

Figure 3: Impacts of minority class $\tilde{l}$ (left) and classifier scale $w$ (right) on the density of Local Outlier Factor (LOF). The other parameters are fixed: $w = 2.0$ (left) and $\tilde{l} = 0$ (right).

tively (see Section C for details on the categorization). The ordinal minority scores are then coupled with the associated data samples to yield a paired dataset $(\boldsymbol{x}_0^{(1)}, \tilde{l}^{(1)}), \ldots, (\boldsymbol{x}_0^{(N)}, \tilde{l}^{(N)})$ which is subsequently used for training a (noise-conditioned) classifier $p_{\boldsymbol{\psi}}(\tilde{l}|\boldsymbol{x}_t)$ that predicts $\tilde{l}$ for input $\boldsymbol{x}_t$ (perturbed via Eq. (1)). After training, we blend the given score model with the log-gradient of the classifier as in Eq. (5) to yield a modified score:

$$\hat{\boldsymbol{s}}_{\boldsymbol{\theta}}(\boldsymbol{x}_t, t, \tilde{l}) := \boldsymbol{s}_{\boldsymbol{\theta}}^*(\boldsymbol{x}_t, t) + w\nabla_{\boldsymbol{x}_t} \log p_{\boldsymbol{\psi}}(\tilde{l} \mid \boldsymbol{x}_t), \tag{8}$$

where $\boldsymbol{\psi}$ indicates parameterization for the classifier, and $w$ is a scaling factor for the guidance; see Figure 3 for details on its impact. Incorporating this mixed score into the sampling process

---

[3]We also explored another natural strategy that concerns sampling from $\hat{q}(\boldsymbol{x}_0) \propto q(\boldsymbol{x}_0)l(\boldsymbol{x}_0; \boldsymbol{s}_{\boldsymbol{\theta}})$. See Section D.5 for details.

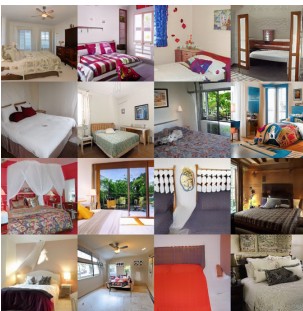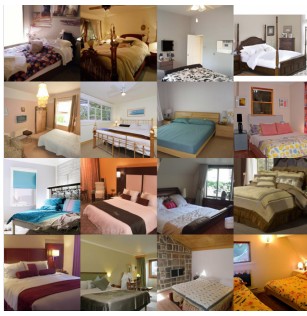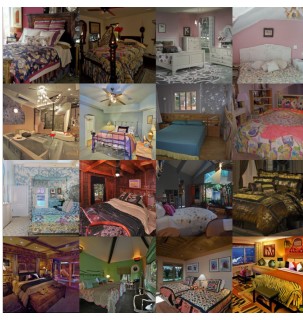

Figure 4: Sample comparison on LSUN-Bedrooms. Generated samples by StyleGAN (Karras et al., 2019) (left), ADM (Dhariwal & Nichol, 2021) with ancestral sampling (middle), and minority guidance (right) are exhibited. We use the same random seed for the diffusion-based samplers.

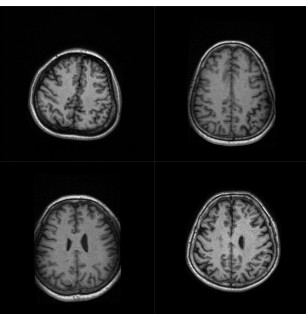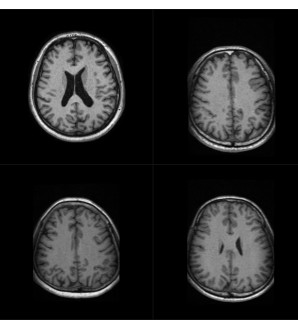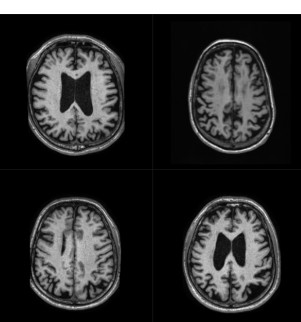

Figure 5: Sample comparison on the brain MRI data. Generated samples by StyleGAN2-ADA (Karras et al., 2020) (left), ADM (Dhariwal & Nichol, 2021) with ancestral sampling (middle), and minority guidance (right) are exhibited. We use the same seed for the diffusion-based samplers.

(in Eq. (3)) then enables conditional generation w.r.t. $\hat{p}_{\boldsymbol{\theta}}(\boldsymbol{x}_t|\tilde{l}) \propto p_{\boldsymbol{\theta}}(\boldsymbol{x}_t)p_{\boldsymbol{\psi}}\big(\tilde{l}|\boldsymbol{x}_t\big)^w$. We call our technique *minority guidance*, as it gives guidance w.r.t. minority score in the generation process.

Generating unique-featured samples is now immediate with minority guidance, e.g., by conditioning an arbitrarily high $\tilde{l}$ with a properly chosen $w$ to incorporate the strength of the associated class $\tilde{l}$. Even more, our sampler enables a *free* control of the uniqueness of features for generated samples, which has never been offered in literature so far to our best knowledge. This uniqueness-controllability could be particularly instrumental in many practical scenarios including medical imaging; see Figure 14 in Section D.7 for one such instance. We found that minority guidance is not confined to the unconditional setting we explored thus far, and can be extended to the class-conditional context for producing low-density samples w.r.t. a specific class. See Section B for details on the extension. This broad applicability that encompasses both unconditional and conditional settings is a key advantage of our approach compared to existing methods for minority generation, which are limited to supervised (i.e., annotation-available) settings only (Yu et al., 2020; Lin et al., 2022; Sehwag et al., 2022).

## 4 EXPERIMENTS

In this section, we provide empirical demonstrations that validate our proposals and arguments through the image generation task, considering both unconditional and class-conditional scenarios. To this end, we first clarify the setup used in our demonstrations (see Section C for more details) and then provide results for our approach in comparison with existing frameworks. A comprehensive collection of our experimental results can be found in Section E.

### 4.1 SETUP

**Datasets.** Our experiments are conducted on six real benchmarks: four unconditional and two class-conditional datasets. For the unconditional settings, we employ CelebA $64^2$ (Liu et al., 2015),

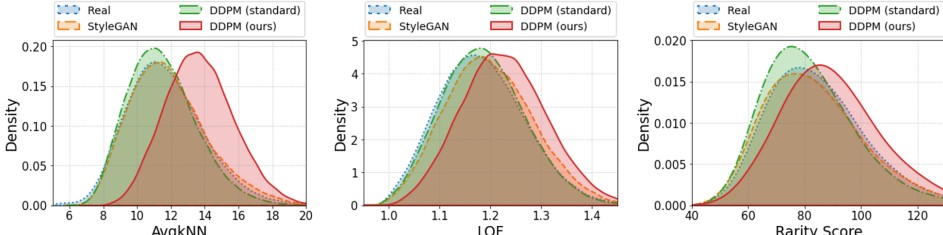

Figure 6: Comparison of neighborhood density on LSUN-Bedrooms. "Real" refers to the training set of LSUN-Bedrooms. "DDPM (standard)" indicates ADM (Dhariwal & Nichol, 2021) with ancestral sampling, while "DDPM (ours)" is ours. The higher values, the less likely samples for all measures.

CIFAR-10 (Krizhevsky et al., 2009), and LSUN-Bedrooms $256^2$ (Yu et al., 2015). In addition to the three unconditional natural image benchmarks, we employ a medical dataset to demonstrate the practical importance of our work. Specifically, we consider an in-house (IRB-approved) brain MRI dataset containing 13,640 axial slice images, where minorities are instances that exhibit degenerative disease like cerebral atrophy. The MRI images are standard 3T T1-weighted with $256^2$ resolution. We use ImageNet $64^2$ and $256^2$ (Deng et al., 2009) for the class-conditional generation. To further investigate the performance on long-tailed data, we additionally incorporate a famous benchmark specifically manipulated to exhibit long-tailed characteristics: CIFAR-10-LT (Cao et al., 2019).

**Baselines.** We consider three prominent GAN-based frameworks for the unconditional cases: (i) BigGAN (Brock et al., 2019); (ii) StyleGAN (Karras et al., 2019); (iii) StyleGAN2-ADA (Karras et al., 2020). In addition to the GAN baselines, we employ a standard sampler for diffusion models, ancestral sampling (a.k.a. the DDPM sampling) (Ho et al., 2020), which we implement with the pretrained diffusion models used in each benchmarks (e.g., ADM (Dhariwal & Nichol, 2021) on CelebA). The BigGAN models are employed for CelebA and CIFAR-10, and we use StyleGAN for LSUN-Bedrooms, while StyleGAN2-ADA is compared in our brain MRI experiments. The DDPM sampling is employed in all three unconditional datasets. To explore potential benefits that may come from the use of minority labels, we incorporate an additional baseline on the CelebA dataset, which conditionally generates minority instances using the classifier guidance and minority annotations available in the data (e.g., "Eyeglasses") (Yu et al., 2020). We also incorporate an unconditional version of Sehwag et al. (2022) on CelebA, which we specifically tailor the original method (developed for class-conditional datasets) for the use in unconditional data. For LSUN-Bedrooms, we additionally consider LDM (Rombach et al., 2022), a famous latent-based framework extensively used in various applications. For the class-conditional ImageNet, we consider two baselines, the standard DDPM sampler and Sehwag et al. (2022). On CIFAR-10-LT, our approach is compared with Qin et al. (2023) and ancestral sampling.

**Evaluation metrics.** For the purpose of evaluating the generation capability of low-density samples, we incorporate three distinct measures for describing the density of neighborhoods: (i) Average k-Nearest Neighbor (AvgkNN); (ii) Local Outlier Factor (LOF) (Breunig et al., 2000); (iii) Rarity Score (Han et al., 2022). For all three measures, a higher value indicates that a given sample is less likely than its neighboring samples (Sehwag et al., 2022; Han et al., 2022). Additionally to investigate fidelity and diversity of generated samples, we employ a set of quantitative metrics: (i) Clean Fréchet Inception Distance (cFID) (Parmar et al., 2022); (ii) Spatial FID (sFID) (Nash et al., 2021); (iii) Improved Precision & Recall (Kynkäänniemi et al., 2019). Since we are interested in producing high-quality diverse samples close to real low-density data, we use the most unique samples (e.g., yielding the highest AvgkNNs) as baseline real data for computing these quality metrics.

## 4.2 RESULTS

**Validation of the roles of $\tilde{l}$ and $w$.** Figure 3 illustrates the impacts of the control parameters of minority guidance on likelihood distributions of generated samples. Observe in the left plot that increasing the minority class $\tilde{l}$ yields the LOF density shifting toward high-LOF (i.e., low-likelihood) regions, demonstrating the capability of our sampler to encourage generations of minorities. On the other hand, the right plot of Figure 3 exhibits that an increase of the classifier scale $w$ makes the density squeezing toward high likelihood regions. This showcases the controllability of minority

| Method | cFID | sFID | Prec | Rec | Method | cFID | sFID | Prec | Rec |
|---|---|---|---|---|---|---|---|---|---|
| **CelebA 64×64** | | | | | **LSUN Bedrooms 256×256** | | | | |
| ADM | 75.05 | 16.73 | **0.97** | 0.23 | ADM | 62.83 | 7.49 | 0.89 | **0.16** |
| ADM-ML | 51.66 | 13.04 | 0.94 | 0.31 | LDM | 63.61 | 7.19 | **0.90** | 0.13 |
| BigGAN | 80.41 | 16.51 | **0.97** | 0.19 | StyleGAN | 56.45 | **5.96** | 0.87 | 0.13 |
| Sehwag et al. | 27.97 | 10.17 | 0.82 | **0.42** | Ours | **41.52** | 6.67 | 0.87 | 0.10 |
| Ours | **26.98** | **8.23** | 0.89 | 0.34 | | | | | |
| **ImageNet 256×256** | | | | | **Brain MRI 256×256** | | | | |
| ADM | 17.41 | 12.85 | **0.87** | 0.35 | ADM | 21.71 | 14.30 | 0.82 | 0.37 |
| Sehwag et al. | **13.43** | 9.88 | 0.85 | 0.42 | StyleGAN2a | 22.69 | 23.07 | 0.53 | 0.28 |
| Ours | 13.83 | **7.82** | 0.85 | **0.45** | Ours | **19.73** | **13.36** | **0.84** | **0.47** |

Table 1: Comparison of sample quality and diversity for generating minority samples. "ADM-ML" refers to a classifier-guided sampler implemented on ADM (Dhariwal & Nichol, 2021), which conditions on **M**inority **L**abels. The best results are marked in bold.

guidance on the strength of features associated with the selected minority class (e.g., $\tilde{l} = 0$ in Figure 3), which also aligns with the well-known role of $w$ emphasized in Dhariwal & Nichol (2021). See Figures 15 and 16 (in Section E) for generated samples that visualize these impacts.

**Comparison with the baselines.** Figure 4 compares generated samples on LSUN-Bedrooms. Observe that our sampler produces more exquisite and unique images than the two baseline methods. Notably, it often encourages some monotonous-looking features generated by the standard DDPM sampler to be more novel, e.g., with inclusions of additional objects. We found that such sophisticated attributes are in fact novel features that yield high minority score values in LSUN-Bedrooms (see Figure 2 for instance). Figure 5 exhibits generated samples on the brain MRI benchmark. We see that our sampler are more likely to generate minority instances of the dataset (e.g., containing severe brain atrophy) compared to the baseline methods (see Figure 13 for visualizations of the minorities). We highlight that the minority-generating capability of our sampler persists even on this particular type of data containing distinctive visual aspects, further demonstrating the practical significance of our method. We leave the generated samples on the other datasets in Section E.

Figure 6 compares our focused density measures on LSUN-Bedrooms. Note that minority guidance outperforms the baselines in generating unique samples for all three measures, which corroborates the visual inspections made on Figure 4. We leave the density results on the other datasets in Section E. Table 1 exhibits quality and diversity evaluations on four challenging benchmarks (see Table 7 for the results on the CIFAR-10 settings and ImageNet-64). For baseline real data, we employ the most unique samples that yield the highest AvgkNN values. Additional evaluations using other criteria for collecting baseline data can be found in Section E; see Table 8 therein. Observe that our sampler yields better (or comparable) results than the baselines in all metrics, demonstrating its ability to produce realistic and diverse samples close to the real minorities. For a more robust evaluation on the MRI data containing distinctive features (hence may not be interpretable in the ImageNet-based feature space), we provide evaluations based on a domain-specific feature space that could admit our MRI data; see Section E for details.

## 5 CONCLUSION AND LIMITATIONS

We emphasized that Tweedie-based denoiser is inherently biased to produce high-likelihood majority samples. In light of this, we introduced a novel metric to evaluate the uniqueness of features and developed a sampling technique that can serve to generate novel-featured minority samples. We demonstrated that the proposed sampler greatly improves the capability of producing high-quality minority samples even on high-stake applications like medical imaging.

One disadvantage is that our approach requires the construction of a minority classifier, which is often computationally expensive when the size of a given dataset is huge. Hence, we believe one promising future direction is to push the boundary towards the challenging scenarios that are data-restricted while addressing the computation issue.

## ETHICS STATEMENT

The brain MRI images employed in our experiments were obtained exclusively for in-house purposes and are not publicly accessible. We emphasize that our MRI dataset has been rigorously reviewed and approved by the Institutional Review Board (IRB), ensuring strict compliance with ethical standards. This includes meticulous attention to informed consent procedures and robust data anonymization protocols.

A potential risk due to our sampler is that it could be used maliciously to suppress the generation of minority-featured samples. This could be achieved, for instance, by using low values of $\tilde{l}$ in Eq. (8), i.e., focusing on producing instances containing majority features. It is important to be aware of this risk and to ensure that our proposed framework is used responsibly to promote fairness and inclusivity in generative modeling.

## REPRODUCIBILITY

To ensure the reproducibility of our experiments, we provide a thorough description regarding our four publicly available benchmark real datasets: CelebA (Liu et al., 2015), CIFAR-10 (Krizhevsky et al., 2009), LSUN-Bedrooms (Yu et al., 2015), and ImageNet (Deng et al., 2009). For our in-house brain MRI dataset which is unable to open to the public, we describe in detail on its characteristics and the way of gathering. All these settings including the choices of hyperparameters are presented in Section C. We also provide the average running time of our algorithm together with specific computer configuration details in the same section. Lastly, to facilitate replication, we make our code available in a public repository: `https://github.com/soobin-um/minority-guidance`.

## ACKNOWLEDGMENTS

This work was supported by National Research foundation of Korea (NRF) (No. RS-2023-00262527), by Field-oriented Technology Development Project for Customs Administration through National Research Foundation of Korea (NRF) funded by the Ministry of Science & ICT and Korea Customs Service (No. NRF-2021M3I1A1097938), by the Korea Medical Device Development Fund grant funded by the Korea government (the Ministry of Science and ICT, the Ministry of Trade, Industry and Energy, the Ministry of Health & Welfare, the Ministry of Food and Drug Safety) (Project Number: 1711137899, KMDF_PR_20200901_0015), by Culture, Sports and Tourism R&D Program through the Korea Creative Content Agency grant funded by the Ministry of Culture, Sports and Tourism in 2023, and by Institute for Information & communications Technology Promotion (IITP) grant funded by the Korea government (MSIP) (No. 2019-0-00075 Artificial Intelligence Graduate School Program (KAIST)).

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

# A PROOFS

## A.1 PROOF OF PROPOSITION 1

**Proposition 1.** *Consider the DSM optimization in Eq. (2). Assume that a given noise-conditioned score network $s_{\boldsymbol{\theta}}(\boldsymbol{x}_t, t)$ have enough capacity. Then for each timestep $t$, the optimality of the score network is achievable at:*

$$s_{\boldsymbol{\theta}}^*(\boldsymbol{x}_t, t) = \mathbb{E}_{q_{\alpha_t}(\boldsymbol{x}_0|\boldsymbol{x}_t)}\left[\nabla_{\boldsymbol{x}_t} \log q_{\alpha_t}(\boldsymbol{x}_t \mid \boldsymbol{x}_0)\right].$$

*Proof.* Since the score network $s_{\boldsymbol{\theta}}(\boldsymbol{x}_t, t)$ is assumed to be large enough, it can achieve the global optimum that minimizes the DSM objectives for all timesteps regardless of their weights in Eq. (2) (Song et al., 2020a). Hence, we can safely ignore the weighted sum and focus on the DSM objective for a specific timestep $t$ to derive $s_{\boldsymbol{\theta}}^*(\boldsymbol{x}_t, t)$:

$$s_{\boldsymbol{\theta}}^*(\boldsymbol{x}_t, t) = \arg\min_{s_{\boldsymbol{\theta}} \in \mathcal{S}} \mathbb{E}_{q(\boldsymbol{x}_0)q_{\alpha_t}(\boldsymbol{x}_t|\boldsymbol{x}_0)}[\|s_{\boldsymbol{\theta}}(\boldsymbol{x}_t, t) - \nabla_{\boldsymbol{x}_t} \log q_{\alpha_t}(\boldsymbol{x}_t \mid \boldsymbol{x}_0)\|_2^2], \tag{9}$$

where we consider the optimization over function space $s_{\boldsymbol{\theta}} \in \mathcal{S}$ instead of the parameter space $\boldsymbol{\theta} \in \Theta$ for the ease of analysis. The objective function in Eq. (9) can be rewritten as:

$$\mathbb{E}_{q(\boldsymbol{x}_0)q_{\alpha_t}(\boldsymbol{x}_t|\boldsymbol{x}_0)}[\|s_{\boldsymbol{\theta}}(\boldsymbol{x}_t, t) - \nabla_{\boldsymbol{x}_t} \log q_{\alpha_t}(\boldsymbol{x}_t \mid \boldsymbol{x}_0)\|_2^2]$$
$$= \int \int q(\boldsymbol{x}_0)q_{\alpha_t}(\boldsymbol{x}_t \mid \boldsymbol{x}_0)\|s_{\boldsymbol{\theta}}(\boldsymbol{x}_t, t) - \nabla_{\boldsymbol{x}_t} \log q_{\alpha_t}(\boldsymbol{x}_t \mid \boldsymbol{x}_0)\|_2^2 \ \mathrm{d}\boldsymbol{x}_t \ \mathrm{d}\boldsymbol{x}_0$$
$$= \int \underbrace{\int q_{\alpha_t}(\boldsymbol{x}_t)q_{\alpha_t}(\boldsymbol{x}_0 \mid \boldsymbol{x}_t)\|s_{\boldsymbol{\theta}}(\boldsymbol{x}_t, t) - \nabla_{\boldsymbol{x}_t} \log q_{\alpha_t}(\boldsymbol{x}_t \mid \boldsymbol{x}_0)\|_2^2 \ \mathrm{d}\boldsymbol{x}_0}_{=:\mathcal{L}(s_{\boldsymbol{\theta}}, \boldsymbol{x}_t)} \ \mathrm{d}\boldsymbol{x}_t.$$

Since the objective function in Eq. (9) is a functional and convex w.r.t. $s_{\boldsymbol{\theta}}$, we can apply the Euler-Lagrange equation (Gelfand et al., 2000) to come up with a condition for the optimality:

$$\frac{\partial \mathcal{L}}{\partial s_{\boldsymbol{\theta}}} = 0 \ \Rightarrow \ \int q_{\alpha_t}(\boldsymbol{x}_t)q_{\alpha_t}(\boldsymbol{x}_0 \mid \boldsymbol{x}_t) \cdot 2\left\{s_{\boldsymbol{\theta}}^*(\boldsymbol{x}_t, t) - \nabla_{\boldsymbol{x}_t} \log q_{\alpha_t}(\boldsymbol{x}_t \mid \boldsymbol{x}_0)\right\} \ \mathrm{d}\boldsymbol{x}_0 = 0.$$

Rearranging the RHS of the arrow gives:

$$\int q_{\alpha_t}(\boldsymbol{x}_t)q_{\alpha_t}(\boldsymbol{x}_0 \mid \boldsymbol{x}_t) \cdot 2\left\{s_{\boldsymbol{\theta}}^*(\boldsymbol{x}_t, t) - \nabla_{\boldsymbol{x}_t} \log q_{\alpha_t}(\boldsymbol{x}_t \mid \boldsymbol{x}_0)\right\} \ \mathrm{d}\boldsymbol{x}_0 = 0$$
$$\Rightarrow \ s_{\boldsymbol{\theta}}^*(\boldsymbol{x}_t, t) \int q_{\alpha_t}(\boldsymbol{x}_0 \mid \boldsymbol{x}_t) \ \mathrm{d}\boldsymbol{x}_0 = \int q_{\alpha_t}(\boldsymbol{x}_0 \mid \boldsymbol{x}_t)\nabla_{\boldsymbol{x}_t} \log q_{\alpha_t}(\boldsymbol{x}_t \mid \boldsymbol{x}_0) \ \mathrm{d}\boldsymbol{x}_0$$
$$\Rightarrow \ s_{\boldsymbol{\theta}}^*(\boldsymbol{x}_t, t) = \mathbb{E}_{q_{\alpha_t}(\boldsymbol{x}_0|\boldsymbol{x}_t)}\left[\nabla_{\boldsymbol{x}_t} \log q_{\alpha_t}(\boldsymbol{x}_t \mid \boldsymbol{x}_0)\right].$$

This completes the proof. □

## A.2 PROOF OF COROLLARY 1

**Corollary 1.** *Minority score, when defined with the squared-error distance loss $\|\cdot\|_2^2$, is equivalent to the noise prediction error w.r.t. timestep $t$ up to scaling:*

$$l(\boldsymbol{x}_0; s_{\boldsymbol{\theta}}^*) = \tilde{w}_t \mathbb{E}_{p(\boldsymbol{\epsilon})}\left[\|\boldsymbol{\epsilon}_{\boldsymbol{\theta}}^*(\boldsymbol{x}_t, t) - \boldsymbol{\epsilon}\|_2^2\right],$$

*where $\tilde{w}_t := (1 - \alpha_t)/\alpha_t$.*

*Proof.* We start from the definition of minority score in Eq. (7):

$$l(\boldsymbol{x}_0; s_{\boldsymbol{\theta}}^*) := \mathbb{E}_{q_{\alpha_t}(\boldsymbol{x}_t|\boldsymbol{x}_0)}[d(\boldsymbol{x}_0, \hat{\boldsymbol{x}}_0)].$$

Plugging the squared-error loss and further manipulations then yield:

$$
\begin{aligned}
l(\boldsymbol{x}_0; \boldsymbol{s}_{\boldsymbol{\theta}}^*) &:= \mathbb{E}_{q_{\alpha_t}(\boldsymbol{x}_t|\boldsymbol{x}_0)}[d(\boldsymbol{x}_0, \hat{\boldsymbol{x}}_0)] = \mathbb{E}_{q_{\alpha_t}(\boldsymbol{x}_t|\boldsymbol{x}_0)}[\|\boldsymbol{x}_0 - \hat{\boldsymbol{x}}_0\|_2^2] \\
&= \mathbb{E}_{q_{\alpha_t}(\boldsymbol{x}_t|\boldsymbol{x}_0)}\left[\left\|\boldsymbol{x}_0 - \frac{1}{\sqrt{\alpha_t}}\{\boldsymbol{x}_t - \sqrt{1-\alpha_t}\boldsymbol{\epsilon}_{\theta}^*(\boldsymbol{x}_t, t)\}\right\|_2^2\right] \\
&= \mathbb{E}_{p(\boldsymbol{\epsilon})}\left[\left\|\frac{\sqrt{1-\alpha_t}}{\sqrt{\alpha_t}}\{\boldsymbol{\epsilon}_{\theta}^*(\boldsymbol{x}_t, t) - \boldsymbol{\epsilon}\}\right\|_2^2\right] \\
&= \frac{1-\alpha_t}{\alpha_t}\mathbb{E}_{p(\boldsymbol{\epsilon})}\left[\|\boldsymbol{\epsilon}_{\theta}^*(\boldsymbol{x}_t, t) - \boldsymbol{\epsilon}\|_2^2\right] \\
&= \tilde{w}_t \mathbb{E}_{p(\boldsymbol{\epsilon})}\left[\|\boldsymbol{\epsilon}_{\theta}^*(\boldsymbol{x}_t, t) - \boldsymbol{\epsilon}\|_2^2\right],
\end{aligned}
\tag{10}
$$

where $\tilde{w}_t := (1-\alpha_t)/\alpha_t$. The second equality is due to the Tweedie's formula (i.e., Eq. (4)) together with the noise-predicting expression $\boldsymbol{s}_{\boldsymbol{\theta}}^*(\boldsymbol{x}_t, t) = -\boldsymbol{\epsilon}_{\theta}^*(\boldsymbol{x}_t, t)/\sqrt{1-\alpha_t}$, and the third equality follows from the DDPM forward process in Eq. (1)). This completes the proof. $\qquad\square$

## B  EXTENSION TO CLASS-CONDITIONAL SETTINGS

We continue from Section 3 of the main paper and explore how to extend our proposed framework toward class-conditional settings. To this end, we first provide a brief background on conditional diffusion models that will be used throughout this section. After that, we elaborate extensions of our metric and guided sampler to our interested class-conditional context.

### B.1  BACKGROUND ON CONDITIONAL DIFFUSION MODELS

Consider joint density $q(\boldsymbol{x}_0, \boldsymbol{c}) = q(\boldsymbol{c})q(\boldsymbol{x}_0|\boldsymbol{c})$ where $\boldsymbol{c}$ is a random variable (or vector) that indicates a specific condition w.r.t. data (e.g., "Water tower" in ImageNet (Deng et al., 2009)). The reverse process to model conditional density $q(\boldsymbol{x}_0|\boldsymbol{c})$ can be defined as $p_{\boldsymbol{\theta}}(\boldsymbol{x}_{t-1}|\boldsymbol{x}_t, \boldsymbol{c}) := \mathcal{N}(\boldsymbol{x}_{t-1}; \boldsymbol{\mu}_{\boldsymbol{\theta}}(\boldsymbol{x}_t, t, \boldsymbol{c}), \beta_t \boldsymbol{I})$ where $\{\beta_t\}_{t=1}^T$ is the same noise schedule as in Section 2. Similar to the unconditional setting, we can express the mean of the reverse conditional density by employing a score network: $\boldsymbol{\mu}_{\boldsymbol{\theta}}(\boldsymbol{x}_t, t, \boldsymbol{c}) = \frac{1}{\sqrt{1-\beta_t}}(\boldsymbol{x}_t + \beta_t \boldsymbol{s}_{\boldsymbol{\theta}}(\boldsymbol{x}_t, t, \boldsymbol{c}))$. The score network is designed to approximate class-conditional score function $\nabla_{\boldsymbol{x}_t} \log q_{\alpha_t}(\boldsymbol{x}_t|\boldsymbol{c})$ and therefore parameterizable as $\boldsymbol{s}_{\boldsymbol{\theta}}(\boldsymbol{x}_t, t, \boldsymbol{c}) := \nabla_{\boldsymbol{x}_t} \log p_{\boldsymbol{\theta}}(\boldsymbol{x}_t|\boldsymbol{c})$. The conditional score network can be trained with a variant of denoising score matching similar to the unconditional case:

$$
\min_{\boldsymbol{\theta}} \sum_{t=1}^T w_t \mathbb{E}_{q(\boldsymbol{c})q(\boldsymbol{x}_0|\boldsymbol{c})q_{\alpha_t}(\boldsymbol{x}_t|\boldsymbol{x}_0)}[\|\boldsymbol{s}_{\boldsymbol{\theta}}(\boldsymbol{x}_t, t, \boldsymbol{c}) - \nabla_{\boldsymbol{x}_t} \log q_{\alpha_t}(\boldsymbol{x}_t \mid \boldsymbol{x}_0)\|_2^2].
\tag{11}
$$

where $w_t := 1 - \alpha_t$. Given the optimal score network $\boldsymbol{s}_{\boldsymbol{\theta}}^*(\boldsymbol{x}_t, t, \boldsymbol{c})$ obtained via Eq. (11), the same ancestral sampling can be used for producing samples from $p_{\boldsymbol{\theta}}(\boldsymbol{x}_0|\boldsymbol{c}) \approx q(\boldsymbol{x}_0|\boldsymbol{c})$:

$$
\boldsymbol{x}_{t-1} = \frac{1}{\sqrt{1-\beta_t}}(\boldsymbol{x}_t + \beta_t \boldsymbol{s}_{\boldsymbol{\theta}}^*(\boldsymbol{x}_t, t, \boldsymbol{c})) + \beta_t \boldsymbol{z}, \quad \boldsymbol{z} \sim \mathcal{N}(\boldsymbol{0}, \boldsymbol{I}).
\tag{12}
$$

### B.2  EXTENSION OF MINORITY SCORE

Our first step is extending Tweedie's formula to the class-conditional setting that we described in Section B.1. Let us consider labeled sample $(\boldsymbol{x}_0, \boldsymbol{c}) \sim q(\boldsymbol{x}_0, \boldsymbol{c})$ and its noised version $\boldsymbol{x}_t \sim q_{\alpha_t}(\boldsymbol{x}_t|\boldsymbol{x}_0)$. For denoising, one can naturally think of using a variant of Tweedie's formula by replacing $\boldsymbol{s}_{\boldsymbol{\theta}}^*(\boldsymbol{x}_t, t)$ in Eq. (6) with $\boldsymbol{s}_{\boldsymbol{\theta}}^*(\boldsymbol{x}_t, t, \boldsymbol{c})$:

$$
\hat{\boldsymbol{x}}_0(\boldsymbol{c}) := \frac{1}{\sqrt{\alpha_t}}\left(\boldsymbol{x}_t + (1-\alpha_t)\boldsymbol{s}_{\boldsymbol{\theta}}^*(\boldsymbol{x}_t, t, \boldsymbol{c})\right).
\tag{13}
$$

In fact, this natural extension yields the posterior mean of the clean sample $\mathbb{E}[\boldsymbol{x}_0|\boldsymbol{x}_t, \boldsymbol{c}]$; see the following proposition for a formal description and the proof.

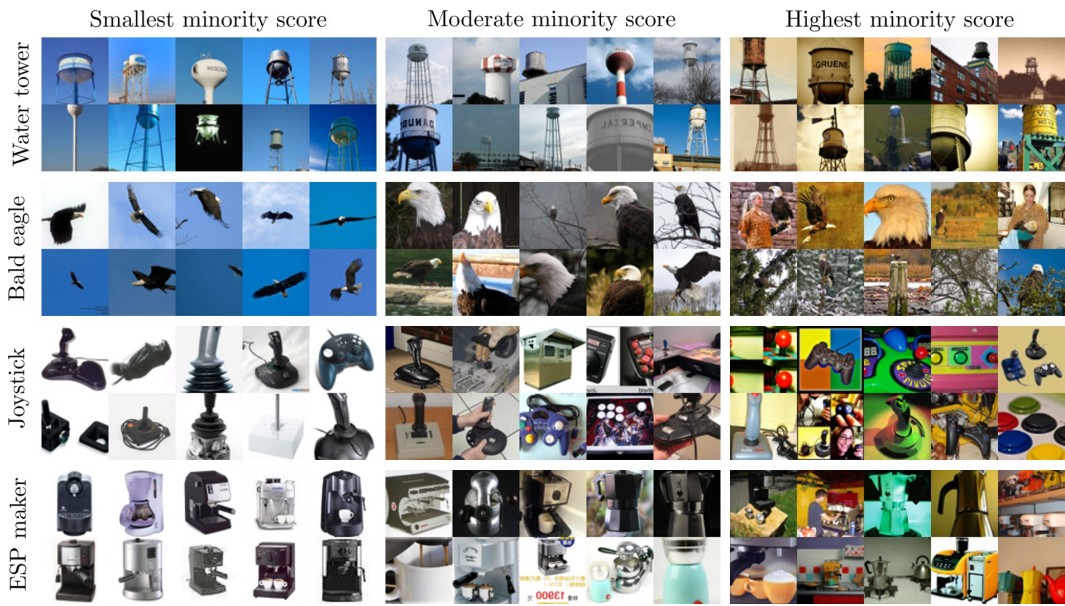

Figure 7: Real samples with the smallest (left column), moderate (middle column), and the highest (right column) minority score. An extended version of minority score (i.e., Eq. (16)) is used on the class-conditional ImageNet $64 \times 64$. We employ Learned Perceptual Image Patch Similarity (Zhang et al., 2018) for $d(\cdot, \cdot)$ in Eq. (16). The considered classes are as follows: water tower (top row), bald eagle (top-middle row), joystick (middle-bottom row), and espresso maker (bottom row).

**Proposition 2.** *Suppose that the optimal score network $s_{\boldsymbol{\theta}}^*(\boldsymbol{x}_t, t, \boldsymbol{c})$ trained via the DSM optimization in Eq. (11) is available. Then for a given labeled sample $(\boldsymbol{x}_0, \boldsymbol{c}) \sim q(\boldsymbol{x}_0, \boldsymbol{c})$ and its noised version $\boldsymbol{x}_t \sim q_{\alpha_t}(\boldsymbol{x}_t|\boldsymbol{x}_0)$, the posterior mean $\mathbb{E}[\boldsymbol{x}_0|\boldsymbol{x}_t, \boldsymbol{c}]$ is achievable at:*

$$\hat{\boldsymbol{x}}_0(\boldsymbol{c}) := \mathbb{E}[\boldsymbol{x}_0|\boldsymbol{x}_t, \boldsymbol{c}] = \frac{1}{\sqrt{\alpha_t}} \left( \boldsymbol{x}_t + (1 - \alpha_t) s_{\boldsymbol{\theta}}^*(\boldsymbol{x}_t, t, \boldsymbol{c}) \right).$$

*Proof.* As a first step, we obtain an analytic form of the optimal score network $s_{\boldsymbol{\theta}}^*(\boldsymbol{x}_t, t, \boldsymbol{c})$ via the following lemma which is in fact an extension of Proposition 1 into the class-conditional context:

**Lemma 1.** *Consider the DSM optimization in Eq. (11). Assume that a given noise-conditioned score network $s_{\boldsymbol{\theta}}(\boldsymbol{x}_t, t, \boldsymbol{c})$ have enough capacity. Then for each timestep $t$ and a given condition $\boldsymbol{c}$, the optimality of the score network is achievable at:*

$$s_{\boldsymbol{\theta}}^*(\boldsymbol{x}_t, t, \boldsymbol{c}) = \mathbb{E}_{q_{\alpha_t}(\boldsymbol{x}_0|\boldsymbol{x}_t, \boldsymbol{c})} \left[ \nabla_{\boldsymbol{x}_t} \log q_{\alpha_t}(\boldsymbol{x}_t \mid \boldsymbol{x}_0) \right]. \tag{14}$$

*Proof.* The proof techniques are essentially the same as the ones used in Proposition 1. Since we assume the score network $s_{\boldsymbol{\theta}}(\boldsymbol{x}_t, t, \boldsymbol{c})$ to have enough capacity, the global optimal point for all timesteps and classes is achievable regardless of $\{w_t\}_{t=1}^T$ in Eq. (11). Therefore, we can focus on the DSM optimization w.r.t. a specific timestep $t$ and a particular class $\boldsymbol{c}$ to obtain $s_{\boldsymbol{\theta}}^*(\boldsymbol{x}_t, t, \boldsymbol{c})$:

$$s_{\boldsymbol{\theta}}^*(\boldsymbol{x}_t, t, \boldsymbol{c}) = \arg \min_{s_{\boldsymbol{\theta}} \in \mathcal{S}} \mathbb{E}_{q(\boldsymbol{x}_0|\boldsymbol{c})q_{\alpha_t}(\boldsymbol{x}_t|\boldsymbol{x}_0)}[\|s_{\boldsymbol{\theta}}(\boldsymbol{x}_t, t, \boldsymbol{c}) - \nabla_{\boldsymbol{x}_t} \log q_{\alpha_t}(\boldsymbol{x}_t \mid \boldsymbol{x}_0)\|_2^2]. \tag{15}$$

Note that we consider the optimization over function space $s_{\boldsymbol{\theta}} \in \mathcal{S}$ rather than the parameter space $\boldsymbol{\theta} \in \Theta$ for simplicity. Rewriting the objective function in Eq. (15) gives:

$$\mathbb{E}_{q(\boldsymbol{x}_0|\boldsymbol{c})q_{\alpha_t}(\boldsymbol{x}_t|\boldsymbol{x}_0)}[\|s_{\boldsymbol{\theta}}(\boldsymbol{x}_t, t, \boldsymbol{c}) - \nabla_{\boldsymbol{x}_t} \log q_{\alpha_t}(\boldsymbol{x}_t \mid \boldsymbol{x}_0)\|_2^2]$$

$$= \int \int q(\boldsymbol{x}_0 \mid \boldsymbol{c}) q_{\alpha_t}(\boldsymbol{x}_t \mid \boldsymbol{x}_0) \|s_{\boldsymbol{\theta}}(\boldsymbol{x}_t, t, \boldsymbol{c}) - \nabla_{\boldsymbol{x}_t} \log q_{\alpha_t}(\boldsymbol{x}_t \mid \boldsymbol{x}_0)\|_2^2 \ \mathrm{d}\boldsymbol{x}_t \ \mathrm{d}\boldsymbol{x}_0$$

$$= \int \underbrace{\int q_{\alpha_t}(\boldsymbol{x}_t \mid \boldsymbol{c}) q_{\alpha_t}(\boldsymbol{x}_0 \mid \boldsymbol{x}_t, \boldsymbol{c}) \|s_{\boldsymbol{\theta}}(\boldsymbol{x}_t, t, \boldsymbol{c}) - \nabla_{\boldsymbol{x}_t} \log q_{\alpha_t}(\boldsymbol{x}_t \mid \boldsymbol{x}_0)\|_2^2 \ \mathrm{d}\boldsymbol{x}_0}_{=: \mathcal{L}(s_{\boldsymbol{\theta}}, \boldsymbol{x}_t)} \ \mathrm{d}\boldsymbol{x}_t.$$

As in Proposition 1, we employ the Euler-Lagrange equation for $\mathcal{L}(s_{\boldsymbol{\theta}}, \boldsymbol{x}_t)$ to come up with a condition for the optimality:

$$\frac{\partial \mathcal{L}}{\partial \boldsymbol{s_{\theta}}} = 0 \quad \Rightarrow \quad \int q_{\alpha_t}(\boldsymbol{x}_t \mid \boldsymbol{c}) q_{\alpha_t}(\boldsymbol{x}_0 \mid \boldsymbol{x}_t, \boldsymbol{c}) \cdot 2 \left\{ \boldsymbol{s}_{\boldsymbol{\theta}}^*(\boldsymbol{x}_t, t, \boldsymbol{c}) - \nabla_{\boldsymbol{x}_t} \log q_{\alpha_t}(\boldsymbol{x}_t \mid \boldsymbol{x}_0) \right\} \, \mathrm{d}\boldsymbol{x}_0 = 0.$$

The RHS of the arrow can be rewritten as:

$$\int q_{\alpha_t}(\boldsymbol{x}_t \mid \boldsymbol{c}) q_{\alpha_t}(\boldsymbol{x}_0 \mid \boldsymbol{x}_t, \boldsymbol{c}) \cdot 2 \left\{ \boldsymbol{s}_{\boldsymbol{\theta}}^*(\boldsymbol{x}_t, t, \boldsymbol{c}) - \nabla_{\boldsymbol{x}_t} \log q_{\alpha_t}(\boldsymbol{x}_t \mid \boldsymbol{x}_0) \right\} \, \mathrm{d}\boldsymbol{x}_0 = 0$$

$$\Rightarrow \boldsymbol{s}_{\boldsymbol{\theta}}^*(\boldsymbol{x}_t, t, \boldsymbol{c}) \int q_{\alpha_t}(\boldsymbol{x}_0 \mid \boldsymbol{x}_t, \boldsymbol{c}) \, \mathrm{d}\boldsymbol{x}_0 = \int q_{\alpha_t}(\boldsymbol{x}_0 \mid \boldsymbol{x}_t, \boldsymbol{c}) \nabla_{\boldsymbol{x}_t} \log q_{\alpha_t}(\boldsymbol{x}_t \mid \boldsymbol{x}_0) \, \mathrm{d}\boldsymbol{x}_0$$

$$\Rightarrow \boldsymbol{s}_{\boldsymbol{\theta}}^*(\boldsymbol{x}_t, t, \boldsymbol{c}) = \mathbb{E}_{q_{\alpha_t}(\boldsymbol{x}_0 \mid \boldsymbol{x}_t, \boldsymbol{c})} \left[ \nabla_{\boldsymbol{x}_t} \log q_{\alpha_t}(\boldsymbol{x}_t \mid \boldsymbol{x}_0) \right].$$

This completes the proof of Lemma 1. $\qquad \square$

The remaining part is straightforward. Plugging the optimal score formula (i.e., Eq. (14)) into the RHS of Eq. (13) yields:

$$\frac{1}{\sqrt{\alpha_t}} \left( \boldsymbol{x}_t + (1 - \alpha_t) \boldsymbol{s}_{\boldsymbol{\theta}}^*(\boldsymbol{x}_t, t, \boldsymbol{c}) \right)$$

$$= \frac{1}{\sqrt{\alpha_t}} \left\{ \boldsymbol{x}_t + (1 - \alpha_t) \mathbb{E}_{q_{\alpha_t}(\boldsymbol{x}_0 \mid \boldsymbol{x}_t, \boldsymbol{c})} \left[ \nabla_{\boldsymbol{x}_t} \log q_{\alpha_t}(\boldsymbol{x}_t \mid \boldsymbol{x}_0) \right] \right\}$$

$$= \frac{1}{\sqrt{\alpha_t}} \left\{ \boldsymbol{x}_t + (1 - \alpha_t) \mathbb{E}_{q_{\alpha_t}(\boldsymbol{x}_0 \mid \boldsymbol{x}_t, \boldsymbol{c})} \left[ -\frac{\boldsymbol{x}_t - \sqrt{\alpha_t} \boldsymbol{x}_0}{1 - \alpha_t} \right] \right\}$$

$$= \mathbb{E}_{q_{\alpha_t}(\boldsymbol{x}_0 \mid \boldsymbol{x}_t, \boldsymbol{c})} \left[ \boldsymbol{x}_0 \right]$$

$$= \mathbb{E} \left[ \boldsymbol{x}_0 \mid \boldsymbol{x}_t, \boldsymbol{c} \right].$$

This completes the proof of Proposition 2. Note that when considering the case where $\boldsymbol{c}$ applies to all data, this also serves as a formal proof of Eq. (4), confirming that Tweedie's formula yields a posterior mean (Chung et al., 2022a). $\qquad \square$

Then by employing the same intuition as in the unconditional setting, we develop a variant of minority score tailored for the class-conditional context:

$$l(\boldsymbol{x}_0, \boldsymbol{c}; \boldsymbol{s}_{\boldsymbol{\theta}}^*) := \mathbb{E}_{q_{\alpha_t}(\boldsymbol{x}_t \mid \boldsymbol{x}_0)}[d(\boldsymbol{x}_0, \hat{\boldsymbol{x}}_0(\boldsymbol{c}))], \tag{16}$$

where $\boldsymbol{s}_{\boldsymbol{\theta}}^*$ now indicates the class-conditioned score model $\boldsymbol{s}_{\boldsymbol{\theta}}^*(\boldsymbol{x}_t, t, \boldsymbol{c})$. Notice that when geared with this conditional version, one can figure out whether a given sample $\boldsymbol{x}_0$ is minority *regarding a specific class* $\boldsymbol{c}$, thus being capable to serve as an outlier metric in the class-conditional context. See Figure 7 for visualization of the effectiveness of the extended minority score.

## B.3 EXTENSION OF MINORITY GUIDANCE

As in the unconditional case, we employ the classifier guidance (Dhariwal & Nichol, 2021) for imposing the condition on the uniqueness. A distinction here is that now we have to handle two conditioning variables, i.e., $\boldsymbol{c}$ and $l$, which involves careful consideration and coordination to ensure effective guidance and control over the sampling process. Below we describe in detail on our approach to attain this.

Consider labeled dataset $\{(\boldsymbol{x}_0^{(i)}, \boldsymbol{c}^{(i)})\}_{i=1}^N \overset{\text{i.i.d.}}{\sim} q(\boldsymbol{x}_0, \boldsymbol{c})$ and conditional diffusion model $\boldsymbol{s}_{\boldsymbol{\theta}}^*(\boldsymbol{x}_t, t, \boldsymbol{c})$ pretrained on that dataset. Similar to the unconditional case, one can compute minority score for each paired sample via Eq. (16) and get $l^{(1)}, \ldots, l^{(N)}$ where $l^{(i)} := l(\boldsymbol{x}_0^{(i)}, \boldsymbol{c}^{(i)}; \boldsymbol{s}_{\boldsymbol{\theta}}^*), i \in \{1, \ldots, N\}$. For categorizing minority score values, we take an intra-class approach to better capture the uniqueness of features associated with each individual class-condition. Specifically for each class $\boldsymbol{c} \in \mathcal{C}$ (available in the dataset), we first aggregate the associated minority score values $\{l^{(i)}\}_{i \in \mathcal{S}_c}$ where $\mathcal{S}_c := \{i : \boldsymbol{c}^{(i)} = \boldsymbol{c}\}$. We then categorize them by applying $L - 1$ distinct threshold levels where

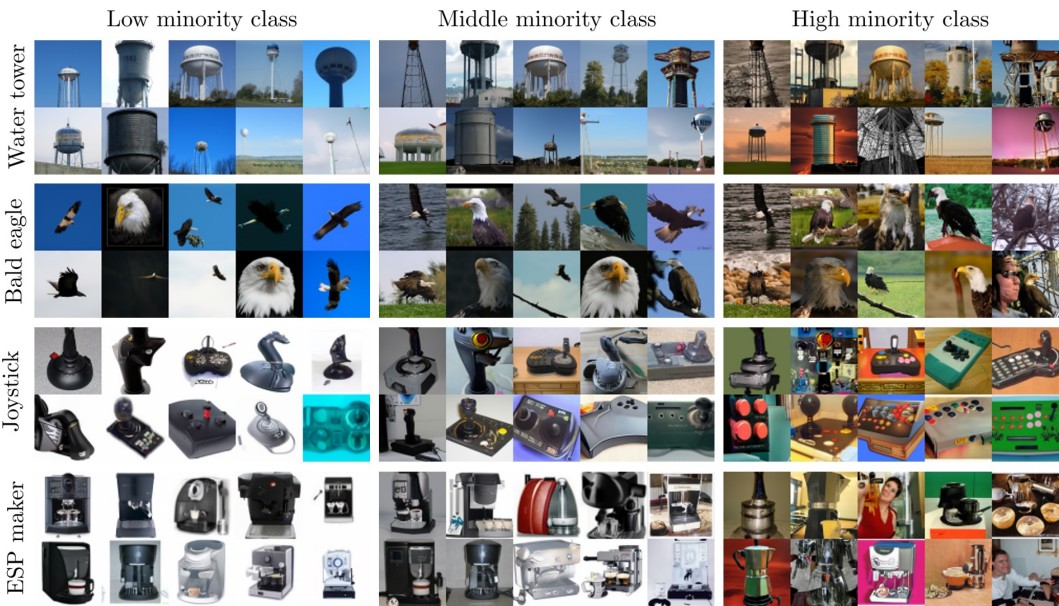

Figure 8: Generated samples by the extended minority guidance over various minority classes. Three different minority classes are considered: $\tilde{l} = 0$ (left column), $\tilde{l} = 12$ (middle column), and $\tilde{l} = 24$ (right column). The same ImageNet classes as in Figure 7 are considered for generation: water tower (top row), bald eagle (top-middle row), joystick (middle-bottom row), and espresso maker (bottom row). We share the same random seed for each row.

$L$ is shared across all classes $\boldsymbol{c} \in \mathcal{C}^4$. This yields the categorized minority scores w.r.t. condition $\boldsymbol{c}$: $\tilde{l}(\boldsymbol{c}^{(1)}), \ldots, \tilde{l}(\boldsymbol{c}^{(N)}) \in \{0, \ldots, L-1\}$ where $\tilde{l}(\boldsymbol{c}) = 0$ and $\tilde{l}(\boldsymbol{c}) = L - 1$ indicate the minority classes w.r.t. the most common and the rarest features regarding condition $\boldsymbol{c}$, respectively. We then couple the ordinal minority scores with the associated paired samples to construct a *tripled* dataset $(\boldsymbol{x}_0^{(1)}, \boldsymbol{c}^{(1)}, \tilde{l}(\boldsymbol{c}^{(1)})), \ldots, (\boldsymbol{x}_0^{(N)}, \boldsymbol{c}^{(N)}, \tilde{l}(\boldsymbol{c}^{(N)}))$. We employ this dataset for training a (class-conditioned) minority predictor $p_{\boldsymbol{\psi}}(\tilde{l}|\boldsymbol{x}_t, \boldsymbol{c})$ that classifies $\tilde{l}$ for perturbed input $\boldsymbol{x}_t$ given class-condition $\boldsymbol{c}$. After training, we mix the given score model $\boldsymbol{s}_{\boldsymbol{\theta}}^*(\boldsymbol{x}_t, t, \boldsymbol{c})$ with the log-gradient of the classifier as in Eq. (5) to yield a modified score:

$$\hat{\boldsymbol{s}}_{\boldsymbol{\theta}}(\boldsymbol{x}_t, t, \boldsymbol{c}, \tilde{l}) := \boldsymbol{s}_{\boldsymbol{\theta}}^*(\boldsymbol{x}_t, t, \boldsymbol{c}) + w \nabla_{\boldsymbol{x}_t} \log p_{\boldsymbol{\psi}}(\tilde{l} \mid \boldsymbol{x}_t, \boldsymbol{c}). \tag{17}$$

Using this blended score as a drop-in replacement of $\boldsymbol{s}_{\boldsymbol{\theta}}^*(\boldsymbol{x}_t, t, \boldsymbol{c})$ in Eq. (12) then implements conditional generation w.r.t. both $\boldsymbol{c}$ and $\tilde{l}$, e.g., according to $\hat{p}_{\boldsymbol{\theta}}(\boldsymbol{x}_t|\boldsymbol{c}, \tilde{l}) \propto p_{\boldsymbol{\theta}}(\boldsymbol{x}_t|\boldsymbol{c}) p_{\boldsymbol{\psi}}(\tilde{l}|\boldsymbol{x}_t, \boldsymbol{c})^w$. For hyperparameters $\tilde{l}$ and $w$ in Eq. (17), we provide visualizations on their respective influences in Figures 8 and 17; see details therein. We empirically found that our extended minority guidance offers better (or comparable) performance to existing class-conditional low-density samplers such as Sehwag et al. (2022). See Sections 4.2 and E for explicit details.

## C  ADDITIONAL DETAILS ON EXPERIMENTAL SETUP

**Categorization strategy.** For the threshold levels to compart the raw minority scores (i.e., ones that are used to turn $l^{(i)}$ into $\tilde{l}^{(i)}$ in Section 3.3), we observed that a naive equally-spaced thresholds yield significant imbalance in the size of classes (e.g., extremely small numbers of samples in highly unique classes), which would then yield negative impacts in the performance of the classifier especially against the small-sized classes. Hence, we resort to splitting the minority scores based on their quantiles. When $L = 10$ for instance, we categorize the minority scores such that $\tilde{l} = 9$ ($\tilde{l} = 0$)

---

[4]We may assign different numbers of thresholds for distinct classes, say $L(\boldsymbol{c})$, especially when a given dataset contains highly imbalanced distributions of classes.

becomes the samples with the top (bottom) $10\%$ of the uniquely-featured samples. For the number of classes $L$, using ones that yield benign numbers of samples per each class (e.g., over 50 samples) chosen based on the size of a given dataset usually offers good performances. See Section D.3 for demonstration. We also found that $L$ can serve as a control knob for balancing the faithfulness of the guidance with the controllability over the uniqueness of features; see Section D.3 for details.

**Medical imaging dataset.** The brain MRI benchmark used in our experiments is a subset of an IRB-approved volumetric dataset that consists of 236,288 2D images with 923 volumes (i.e., images from 923 distinct subjects). All volumes are in the shape of $256 \times 256 \times 256$ cube and standard 3T T1-weighted images with 1mm thickness. The raw (volumetric) dataset has two different data sources, where 482 (out of 923) volumes are from Alzheimer's Disease Neuroimaging Initiative (ADNI) dataset, comprising 271 subjects with probable dementia and 211 subjects with normal cognition. The remaining 441 volumes were collected from a university hospital's voluntary health screening program, with a group of participants that have normal cognition. For the use of this volumetric data in our experiments, we chose 20 specific slice positions (per each volume) that prominently exhibit abnormal features in the dataset (e.g., brain atrophy). We extracted these selected 20 slices from all 652 volumes with normal features, while for the 271 abnormal volumes, we deliberately restricted our selection to only 30 volumes to ensure a low likelihood of encountering the abnormal features. The selected 13,640 slice images are normalized to fall within the range of $[0, 1]$ (per slice).

**Pretrained models.** The backbone diffusion model for the CelebA results is constructed using the architecture and the training setting used for ImageNet $64 \times 64$ in Dhariwal & Nichol (2021)[5]. For the unconditional CIFAR-10 model, we employ the checkpoint provided in Nichol & Dhariwal (2021)[6] without fine-tuning or modification. The pretrained models for LSUN-Bedrooms and ImageNet were taken from Dhariwal & Nichol (2021) and used without modifications. For the brain MRI results, we train the backbone model by employing the same architecture and the setting used for LSUN-Bedrooms in Dhariwal & Nichol (2021).

**Classifiers for minority guidance.** The classifiers used for minority guidance are based on the encoder architecture of U-Net used in Dhariwal & Nichol (2021). Except for LSUN-Bedrooms, we employ all training samples for constructing the minority classifiers (e.g., $N = 50000$ for CIFAR-10). On the other hand, only a $10\%$ of the training samples are used for LSUN-Bedrooms[7]. For the number of minority classes $L$, we take $L = 100$ for the three unconditional natural image datasets and $L = 25$ for ImageNet and CIFAR-10-LT. We use $L = 50$ for the brain MRI experiments.

We construct the minority classifier used on LSUN-Bedrooms by employing a $10\%$ of the training samples provided in a publicly available Kaggle repository[8]. Specifically for the LSUN-Bedrooms and ImageNet classifiers, we employ pretrained feature extractors to encourage efficient training as in (Kim et al., 2022). More precisely, we put $(\boldsymbol{x}_t, t)$ to a feature extractor, and extract the latent $\boldsymbol{z}_t$ of $\boldsymbol{x}_t$ from the last pooling layer of the extractor network. We then feed $(\boldsymbol{z}_t, t)$ to a minority classifier and predict its minority class $\tilde{l}$. For the feature extractors, we employ the pretrained ImageNet classifiers provided in Dhariwal & Nichol (2021) and freeze them during training (as in (Kim et al., 2022)). The classifier architecture for LSUN-Bedrooms is the same as the shallow U-Net used for ImageNet $256 \times 256$ in (Kim et al., 2022). For ImageNet, we employ the shallow U-Net architecture used for CelebA in (Kim et al., 2022) with class-conditioning (to incorporate $\boldsymbol{c}$ in Eq. (17)). The classifier architecture for the brain MRI experiments is the same as the one used in Wolleb et al. (2022). See Table 2 for explicit details.

**Baselines.** The BigGAN model for the CIFAR-10 experiments are trained with the setting provided in the author's official project page[9]. For the CelebA model, we respect the architecture used in Choi et al. (2020)[10] and follow the same training setup as the CIFAR-10 model. We employ StyleGAN for LSUN-Bedrooms using the checkpoint provided in Karras et al. (2019)[11]. To obtain the baseline

---

[5] https://github.com/openai/guided-diffusion

[6] https://github.com/openai/improved-diffusion

[7] We found that the performance of minority guidance is indeed affected by the number of samples employed for constructing the minority classifier. See Section D.4 for details.

[8] https://www.kaggle.com/datasets/jhoward/lsun_bedroom

[9] https://github.com/ajbrock/BigGAN-PyTorch

[10] https://github.com/ermongroup/fairgen

[11] https://github.com/NVlabs/stylegan

| | CelebA | CIFAR-10 | LSUN | ImageNet | Brain MRI |
|---|---|---|---|---|---|
| **Feature extractor** | | | | | |
| Model | Unused | Unused | ADM | ADM | Unused |
| Input shape | ✗ | ✗ | (B,256,256,3) | (B,64,64,3) | ✗ |
| Output shape | ✗ | ✗ | (B,8,8,512) | (B,8,8,512) | ✗ |
| | | | | | |
| **U-Net encoder architecture** | | | | | |
| Input shape | (B,64,64,3) | (B,32,32,3) | (B,8,8,512) | (B,8,8,512) | (B,256,256,1) |
| Output shape | (B,100) | (B,100) | (B,100) | (B,25) | (B,50) |
| Diffusion steps | 1000 | 4000 | 1000 | 1000 | 1000 |
| Noise schedule | cosine | cosine | linear | cosine | linear |
| Channels | 64 | 32 | 128 | 128 | 32 |
| Depth | 2 | 2 | 2 | 4 | 4 |
| Channels multiple | 1,2,3,4 | 1,2,2,4 | 1 | 1 | 1,1,2,2,4,4 |
| Heads channels | 64 | 64 | 64 | 64 | 64 |
| Attention resolution | 32,16,8 | 16,8 | 8 | 8 | 32,16,8 |
| BigGAN up/down | ✓ | ✓ | ✗ | ✗ | ✓ |
| Attention pooling | ✓ | ✓ | ✓ | ✓ | ✓ |
| Class-conditional | ✗ | ✗ | ✗ | ✓ | ✗ |
| | | | | | |
| **Training setup** | | | | | |
| Weight decay | 0.05 | 0.05 | 0.05 | 0.2 | 0.05 |
| Batch size | 256 | 128 | 256 | 1024 | 256 |
| Iterations | 20K | 12K | 60K | 60K | 5K |
| Learning rate | 3e-4 | 3e-4 | 3e-4 | 6e-4 | 3e-4 |
| | | | | | |
| **Sampling** | | | | | |
| Minority class ($\tilde{l}$) | $\mathcal{U}\{90,99\}$ | $\mathcal{U}\{85,99\}$ | 99 | 24 | $\mathcal{U}\{45,49\}$ |
| Classifier scale ($w$) | 4.0 | 8.0 | 3.5 | 2.5 | 1.5 |
| NFE | 250 | 250 | 250 | 250 | 250 |

Table 2: Training and sampling configurations for our proposed sampler, *minority guidance*. "ADM" is the diffusion model framework proposed in Dhariwal & Nichol (2021). "cosine" refers to noise schedule proposed in Nichol & Dhariwal (2021), and "linear" indicates linearly-spaced noise schedule used in the original DDPM (Ho et al., 2020).

StyleGAN2-ADA model for our MRI experiments, we respect the settings provided in the official codebase[12] and train the model by ourselves. The DDPM baseline and Sehwag et al. (2022) share the same pretrained diffusion models as ours. Since there is no open codebase for Sehwag et al. (2022), we implement their method by ourselves by following the descriptions provided in the original paper (Sehwag et al., 2022). More specifically, we employ the same pretrained diffusion model as ours, i.e., the ImageNet checkpoint provided in (Dhariwal & Nichol, 2021). The classifier used in their sampler is imported from Dhariwal & Nichol (2021), and the real-fake discriminator is constructed via the same architecture and the training setting used in Sehwag et al. (2022). For the additional CelebA baseline that uses the classifier guidance targeting minority annotations, we construct a predictor that classifies four minority attributes: (i) "Bald"; (ii) "Eyeglasses"; (iii) "Mustache"; (iv) "Wearing_Hat". During inference time, we produced samples with random combinations of the four attributes (e.g., bald hair yet not wearing glasses) using the classifier guidance. We use the same pretrained model as DDPM and ours for the additional CelebA baseline.

**Evaluation metrics.** We employ implementation from PyOD (Zhao et al., 2019)[13] to compute Average k-Nearest Neighbor (AvgkNN) and Local Outlier Factor (LOF) (Breunig et al., 2000). We respect the conventional choices for the numbers of nearest neighbors for computing the measures:

---

[12]https://github.com/NVlabs/stylegan2-ada-pytorch
[13]https://pyod.readthedocs.io/en/latest/

5 (AvgkNN) and 20 (LOF). As in (Sehwag et al., 2022), the two metrics are computed in the feature space of ResNet-50. We compute Rarity Score (Han et al., 2022) with $k = 5$ using implementation provided in the official project page[14]. To evaluate the three neighborhood measures for the brain MRI dataset whose modality is distinctive (from that of natural images), we employ a random embedding space which has been demonstrated as being particularly instrumental when interpreting data type different from that of natural images (Ulyanov et al., 2018; Naeem et al., 2020). More specifically, we employ a randomly-initialized feature space of ResNet-50. Additionally to the random embedding space, we further investigate the neighborhood measures on an adapted feature space which is constructed by fine-tuning a pretrained DINO (Caron et al., 2021) on our brain MRI dataset. The fine-tuning is performed on ViT-B/16 architecture respecting the same training setting provided in the official codebase[15] yet with decreased learning rate by tenth.

The computations of clean Fréchet Inception Distance (cFID) (Parmar et al., 2022) are due to the official implementation[16]. For computing spatial FID (Nash et al., 2021), we modify the official pytorch FID (Heusel et al., 2017)[17] to use spatial features (i.e., the first 7 channels from the intermediate `mixed_6/conv` feature maps), rather than the standard `pool_3` inception features. The results of Improved Precision & Recall (Kynkäänniemi et al., 2019) are obtained with $k = 5$ based on the official codebase of (Han et al., 2022). For the brain MRI data, we additionally report the quality metrics evaluated in the DINO fine-tuned feature space (see the above paragraph for details on the fine-tuning). In an effort to evaluate the closeness to our focused low-density data, we employ the most unique samples as baseline real data for computing the quality metrics. More precisely, we employ the 10K and 5K real samples yielding the highest AvgkNN values for CelebA and CIFAR-10, respectively. For LSUN-Bedrooms and ImageNet, the most unique 50K samples with the highest AvgkNNs are employed for baseline real data. We employ the most unique 1K samples for constructing the MRI baseline data. All quality metrics are computed with 50K generated samples.

**Computational complexity.** The inference time of our minority guidance is in line with that of classifier guidance by Dhariwal & Nichol (2021), which takes approximately 0.15 seconds per image with 250 NFE in our CIFAR-10 experiments. On the other hand, the baseline ancestral sampling (with the same 250 NFE) spends 0.12 seconds per image on the same dataset, which is relatively cheap due to the absence of the classifier guidance in the sampling process. In addition to the inference time, our sampler requires some time for the construction of a minority classifier, which totals around 23 minutes for the CIFAR-10 dataset. Specifically, 3 minutes are dedicated to the minority-class labeling process (which is described in Section 3.3), while 20 minutes are spent on the classifier training. All these results were obtained using a single A100 GPU.

**Other details.** For the distance measure in minority score (i.e., $d(\cdot, \cdot)$ in Eq. (7)), we employ Learned Perceptual Image Patch Similarity (LPIPS) (Zhang et al., 2018) for all of our experiments; see Section D.1 for empirical evidence that supports this choice. For the perturbation timestep used for computing minority score (i.e., $t$ in Eq. (7)), we employ $t = 0.9T$ for the models pretrained with the cosine schedule (Nichol & Dhariwal, 2021) and $t = 0.6T$ for the ones built with the linear schedule (Ho et al., 2020), where $T$ indicates the total number of steps that a given backbone model is trained with. For instance, we use $t = 900$ on CelebA where the pretrained model is configured with $T = 1000$. We leave in Section D.6 for details on ablating $t$.

For each dataset, the range of minority class for generation (i.e., $\tilde{l}$ in Eq. (8)) is chosen as the one that achieves the best balance among fidelity, diversity, and low-densitiness of generated samples. We swept over $\{1.0, 1.5, 2.0, 2.5, 3.0, 3.5, 4.0, 5.0, 6.0, \ldots, 10.0\}$ for the classifier scale $w$. We employ 250 timesteps to sample from the baseline DDPM and Sehwag et al. (2022) for all the results. For our minority guidance, the same 250 timesteps are mostly employed, yet we occasionally use 100 timesteps for speeding-up sampling (e.g., when conducting ablation studies), since we found that it yields little degradation as reported in Dhariwal & Nichol (2021). See Table 2 for details on the sampling parameters.

---

[14]`https://github.com/hichoe95/Rarity-Score`
[15]`https://github.com/facebookresearch/dino`
[16]`https://github.com/GaParmar/clean-fid`
[17]`https://github.com/mseitzer/pytorch-fid`

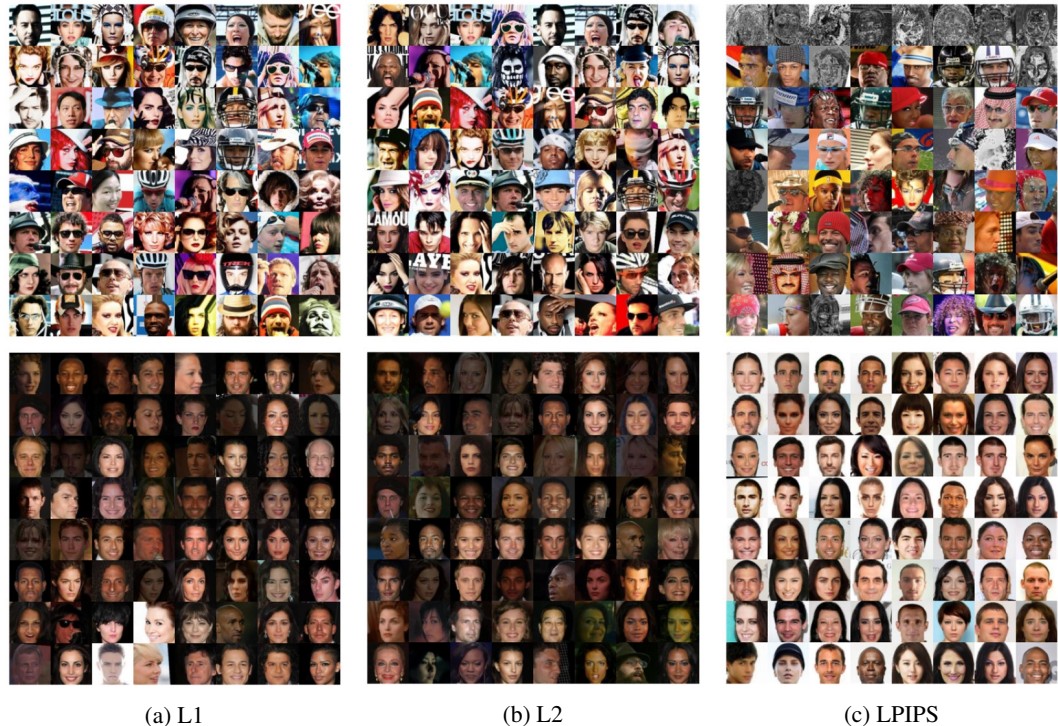

| (a) L1 | (b) L2 | (c) LPIPS |

Figure 9: Performance comparison of minority score with various distance metrics. The most novel (top row) and the most common (bottom row) CelebA real training samples determined by minority scores based on (a) L1; (b) L2; and (c) LPIPS are exhibited.

Our implementation is based on PyTorch (Paszke et al., 2019), and all experiments are performed on twin A100 GPUs. Code is available at `https://github.com/soobin-um/minority-guidance`.

## D ABLATION STUDIES, ANALYSES, AND DISCUSSIONS

We continue from Sections 3 and 4 of the main paper and provide ablation studies and discussions on the proposed approach. We first make an investigation for choosing a proper distance metric that can yield well-performing minority score. We subsequently present additional evidence to reinforce the effectiveness of the minority score as a detector of minority instances. After that, we ablate the number of classes $L$ and investigate its impact on the performance of minority guidance. We also explore the sensitiveness of our algorithm w.r.t. the number of available samples $N$. We then explore another strategy for the minority-focused generation, which is naturally given by our minority score. We further perform an ablation study on $t$, the perturbation timestep for minority score. Lastly, we conduct additional investigations on our in-house brain MRI dataset to highlight the practical significance of our work.

### D.1 THE DISTANCE METRIC FOR MINORITY SCORE

Figure 9 compares the performances of several versions of minority score employing distinct distance metrics. We see that samples containing bright and highly-saturated visual aspects obtain high values from the L1 and L2 versions. This implies that the minority scores with such distances are sensitive to less-semantic information like brightness and saturation. On the other hand, we observe that the minority score

|  | Unique-1K | Common-1K |
|---|---|---|
| L1 | 121.33 | 69.24 |
| L2 | 119.21 | 68.80 |
| LPIPS | 102.66 | 130.57 |

Table 3: Sensitiveness to brightness of the three versions of minority score.

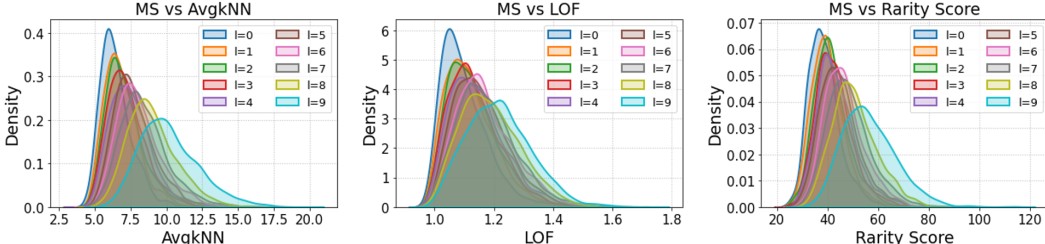

Figure 10: Correlations of minority score with existing outlier metrics. For the CelebA test samples, we compute minority score and the three outlier metrics, and display the correlations between minority score and each outlier metric. A higher value of $l$ indicates a density of more unique samples in terms of minority score. For instance, $l = 9$ denotes a density of instances that yield the top $10\%$ values of minority score. "MS" denotes minority score, our proposed metric. "AvgkNN" refers to Average k-Nearest Neighbor, and "LOF" is Local Outlier Factor (Breunig et al., 2000).

based on LPIPS does not exhibit such sensitiveness,
yielding results that are disentangled with brightness and saturation of images. Moreover, it succeeds in identifying some weird-looking abnormal samples latent in the dataset (see the samples in the first row of the right column) while the other versions fail to do so due to their distraction, e.g., to bright and highly-saturated instances. Table 3 compares the brightness measured on images that are collected using the three versions of minority scores with distinct metrics: L1, L2, and LPIPS. The average brightness values of the most and least unique 1K samples determined via each of the three versions are exhibited. Notice that the minority scores with L1 and L2 distances are sensitive to changes in such less semantic information compared to the one based on LPIPS. We therefore choose LPIPS as the distance metric in computing minority score and use it for all of our experimental results.

## D.2 EFFECTIVENESS OF MINORITY SCORE AS AN OUTLIER METRIC

We argued in the manuscript that minority score is effective for mining minority samples (by giving them high values), and provided empirical evidence for the claim by visualizing samples with various levels of minority score (e.g., in Figure 2). The argument was further strengthened by our experiments that exhibit an improved capability of our sampler (that actually focuses on high-minority-scored instances) for producing low-density samples (see Figure 6 for explicit details). To further validate the success of our metric, we additionally provide empirical evidence of the effectiveness of our metric by conducting quantitative evaluations comparing minority score with other well-established outlier measures, such as Average k-Nearest Neighbor (AvgkNN) and Local Outlier Factor (LOF).

Figure 10 illustrates correlations of minority score with existing well-known outlier metrics. Notice that minority score has positive correlations with other metrics, further demonstrating its ability to reliably capture the uniqueness and minority characteristics of samples.

## D.3 THE NUMBER OF CLASSES $L$ FOR MINORITY GUIDANCE

We first discuss how the number of classes $L$ can be used for balancing between the faithfulness of the guidance and the controllability over the uniqueness of features. We then provide an empirical study that ablates the number of classes $L$ and explores its impact on the performance of minority guidance.

Consider a case where we use a small $L$ for constructing ordinal minority scores. This leads to large numbers of samples per class and therefore helps to build a faithful classifier that well captures class-wise representations. However, if we use too small $L$, then it induces samples containing wide variety of features being flocked into small categorical ranges, thus making it difficult to produce them separately (i.e., losing the controllability). One extreme case that helps understand this is when $L = 1$, where the sampler then becomes just the standard one (that uses Eq. (3)) where we don't have any control over the uniqueness of features. On the other hand, employing large $L$ would

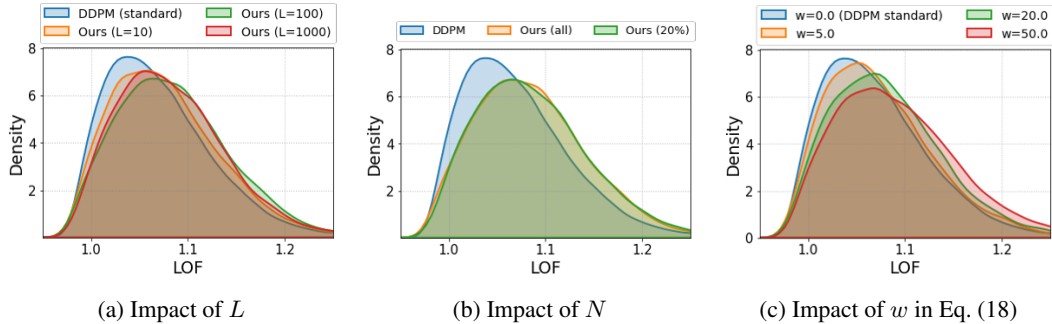

(a) Impact of $L$          (b) Impact of $N$          (c) Impact of $w$ in Eq. (18)

Figure 11: Impact of design choices on the density of Local Outlier Factor (LOF) (Breunig et al., 2000). The influences of three parameters are exhibited: (a) the number of minority classes $L$; (b) the number of available samples $N$; (c) the mixing intensity $w$ in the mixture score approach (i.e., Eq. (18)). The results are obtained on CIFAR-10. The higher LOF, the less likely samples.

make the generation highly controllable w.r.t. the uniqueness of features. However, too large $L$ results in vast numbers of small-sized classes, which makes the classifier hard to capture class-wise representations properly, thereby resulting in a crude predictor that rarely gives meaningful guidance. Considering the other extreme case $L = N$ where the number of classes is the same as the number of samples, the classifier therein would never be able to learn representations over classes well enough for making meaningful guidance for the uniqueness of features. Therefore, we can say that it is important to set a proper $L$ that well balances between the controllability and the faithfulness. Nonetheless, we empirically observed that our minority guidance is robust to the choice of $L$, and a properly selected (yet not that difficult to come up with) $L$ yielding benign numbers of samples for each classes (e.g., containing over 50 samples per class) performs well regardless of $L$, i.e., effectively producing samples with desired features while maintaining realistic sample quality. Below we provide demonstrations that validate the above descriptions.

Figure 11a compares results obtained with a variety of minority classes. Observe that for all considered $L$, our approach improves generative capability of the minority samples over the baseline DDPM sampler, which demonstrates its robustness w.r.t. the number of minority classes $L$. However, we note that $L = 100$ offers better guidance for the minority features than $L = 1000$, yielding a more tilted LOF density toward minority regions (i.e., high LOF values). This validates the role of $L$ as a balancing knob for the faithfulness of the guidance and the controllability over the uniqueness.

### D.4 THE NUMBER OF AVAILABLE SAMPLES $N$

Figure 11b and Table 4 provide empirical results that ablate the number of available samples $N$ specifically on CIFAR-10. Observe that even with limited numbers of samples that are much smaller than the full training set, minority guidance still offers significant improvement in the minority-focused generation over the baseline DDPM sampler. In fact, this benefit is already demonstrated via our experiments on LSUN-Bedrooms presented in Section 4.2 where we only take a 10% of the total training set in obtaining great improvement; see Figures 27 and 20 therein. However, we see degraded diversity compared to the full dataset case, which is reflected in a lower recall (Kynkäänniemi et al., 2019) value; see Table 4 for explicit details.

| Method | Recall |
|---|---|
| Ours (all) | 0.6254 |
| Ours (20%) | 0.5646 |

Table 4: Impact of $N$ on sample diversity. The results are obtained on CIFAR-10 using 10K generated samples for each method.

### D.5 GENERATION FROM $\hat{q}(\boldsymbol{x}_0) \propto q(\boldsymbol{x}_0) l(\boldsymbol{x}_0; \boldsymbol{s}_{\boldsymbol{\theta}})$

Recall our setting considered in Section 3.3 where we have a dataset $\boldsymbol{x}_0^{(1)}, \ldots, \boldsymbol{x}_0^{(N)} \overset{\text{i.i.d.}}{\sim} q(\boldsymbol{x}_0)$, a pretrained diffusion model $\boldsymbol{s}_{\boldsymbol{\theta}}^*(\boldsymbol{x}_t, t)$, and minority score values associated with the dataset $l^{(1)}, \ldots, l^{(N)}$ where $l^{(i)} := l(\boldsymbol{x}_0^{(i)}, \boldsymbol{s}_{\boldsymbol{\theta}}^*), i \in \{1, \ldots, N\}$. Since our interest is to make samples with

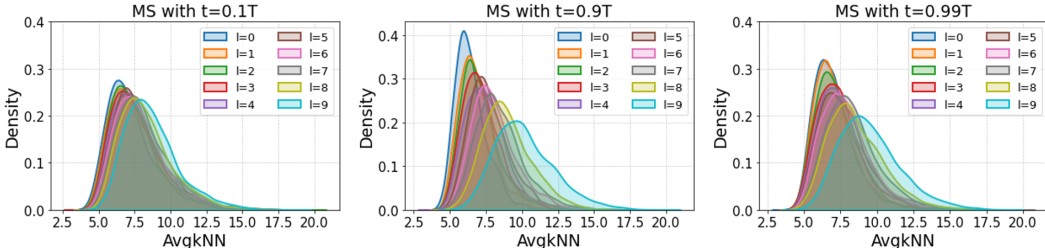

Figure 12: Comparison of minority scores with different choices of the perturbation timestep $t$. For the CelebA test samples, we compute Average k-Nearest Neighbor (AvgkNN) and minority scores with three different timesteps: (i) $t = 0.1T$ (left); (ii) $t = 0.9T$ (middle); (iii) $t = 0.99T$ (right), and exhibit the correlations between AvgkNN and each version of minority score. $T$ indicates the total number of steps that a given backbone model is trained with (e.g., $T = 1000$ for CelebA). The perturbations are based on the cosine noise schedule (Nichol & Dhariwal, 2021). A higher value of $l$ indicates a density of more unique samples based on the respective minority score. For instance, $l = 9$ denotes a density of instances that yield the highest $10\%$ values of minority score. "MS" refers to minority score, our proposed metric. "AvgkNN" refers to Average k-Nearest Neighbor.

high $l^{(i)}$ (i.e., minority instances) more likely to be produced, one can naturally think of sampling from a *blended* distribution $\hat{q}(\boldsymbol{x}_0) \propto q(\boldsymbol{x}_0) l(\boldsymbol{x}_0; \boldsymbol{s}_{\boldsymbol{\theta}}^*)$. However, simulating the generation process w.r.t. this mixed distribution is not that straightforward, since we do not have access to the score functions of its perturbed versions $\{\nabla_{\boldsymbol{x}_t} \log \hat{q}_{\alpha_t}(\boldsymbol{x}_t)\}_{t=1}^T$ where $\hat{q}_{\alpha_t}(\boldsymbol{x}_t) := \int \hat{q}(\boldsymbol{x}_0) q_{\alpha_t}(\boldsymbol{x}_t | \boldsymbol{x}_0) d\boldsymbol{x}_0$.

To circumvent the issue, we take an approximate approach for obtaining the perturbed mixtures. Initially, we take an approximation $\tilde{q}_{\alpha_t}(\boldsymbol{x}_t)$ of the perturbed distribution $\hat{q}_{\alpha_t}(\boldsymbol{x}_t)$ as $\tilde{q}_{\alpha_t}(\boldsymbol{x}_t) \propto q_{\alpha_t}(\boldsymbol{x}_t) l(\boldsymbol{x}_t)$ where $l(\boldsymbol{x}_t)$ denotes minority score w.r.t. a noised instance $\boldsymbol{x}_t$ (describing the uniqueness of $\boldsymbol{x}_t$). This gives $\nabla_{\boldsymbol{x}_t} \log \tilde{q}_{\alpha_t}(\boldsymbol{x}_t) = \boldsymbol{s}_{\boldsymbol{\theta}}^*(\boldsymbol{x}_t, t) + \nabla_{\boldsymbol{x}_t} \log l(\boldsymbol{x}_t)$. Since $l(\boldsymbol{x}_t)$ is intractable, we resort to considering an approximated version $\hat{l}(\boldsymbol{x}_t)$ by computing an empirical average via the use of the minority classifier $p_{\boldsymbol{\psi}}(\tilde{l}|\boldsymbol{x}_t)$:

$$\hat{l}(\boldsymbol{x}_t) := \sum_{i=0}^{L-1} \tau_i \cdot p_{\boldsymbol{\psi}}(\tilde{l} = i \mid \boldsymbol{x}_t),$$

where $\{\tau_i\}_{i=0}^{L-1}$ denote the threshold levels used for categorizing minority score (see Section 3.3 for details). The score function of the perturbed mixture at timestep $t$ then reads:

$$\nabla_{\boldsymbol{x}_t} \log \tilde{q}_{\alpha_t}(\boldsymbol{x}_t) \approx \boldsymbol{s}_{\boldsymbol{\theta}}^*(\boldsymbol{x}_t, t) + w \nabla_{\boldsymbol{x}_t} \log \hat{l}(\boldsymbol{x}_t), \tag{18}$$

where $w$ is a parameter introduced for controlling the intensity of the mixing; see Figure 11c for demonstrations of its impact. We found that using the mixed score as a drop in replacement of $\boldsymbol{s}_{\boldsymbol{\theta}}^*(\boldsymbol{x}_t, t)$ in the standard sampler (i.e., Eq. (3)) improves the capability of diffusion models to generate low-density minority samples.

Figure 11c exhibits the ability of the mixed score approach in Eq. (18) to generate minority samples concerning a variety of $w$. Notice that increasing $w$ yields the LOF density shifting toward high-valued regions (i.e., low-density regions), implying that $w$ plays a similar role as $\tilde{l}$ in minority guidance and therefore can be used for improving the generation capability for minority features. While this alternative sampler based on the mixed distribution $\hat{q}(\boldsymbol{x}_0) \propto q(\boldsymbol{x}_0) l(\boldsymbol{x}_0; \boldsymbol{s}_{\boldsymbol{\theta}}^*)$ yields significant improvement over the standard DDPM sampler (i.e., $w = 0.0$ case), minority guidance still offers better controllability over the uniqueness of features thanks to the collaborative twin knobs $\tilde{l}$ and $w$. Hence, we adopt it as our main gear for tackling the preference of the diffusion models w.r.t. majority features.

### D.6   THE PERTURBATION TIMESTEP $t$ FOR MINORITY SCORE

Figure 12 exhibits performances of minority score with distinct choices of the perturbation timestep $t$. Notice that when using small (or too high) values of $t$, the corresponding minority score does

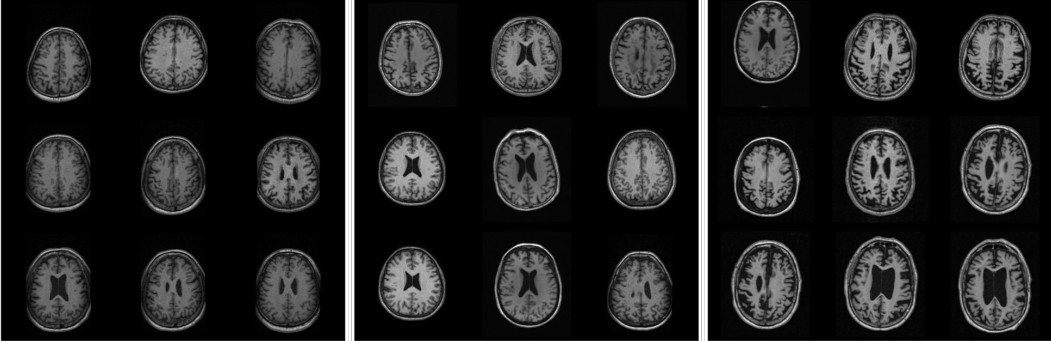

Figure 13: Real samples in our focused brain MRI dataset across minority score values. Data instances that yield the smallest (left), moderate (middle), and the highest (right) minority scores are exhibited herein.

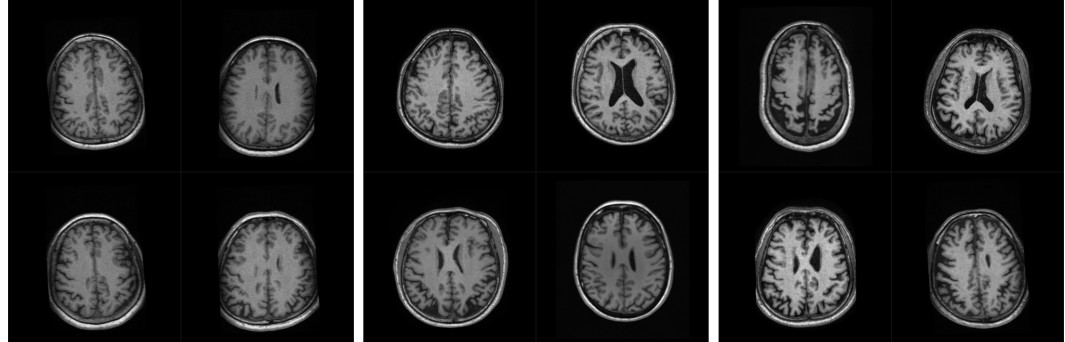

Figure 14: Generated samples by minority guidance on the brain MRI dataset over various minority classes. Three different classes are considered herein: $\tilde{l} = 0$ (left), $\tilde{l} = 35$ (middle), and $\tilde{l} = 49$ (right). The classifier scale is fixed to $w = 2.5$ for all three settings. We share the same random seed for all generations.

not well correlate with Average k-Nearest Neighbor (AvgkNN), meaning its lack of effectiveness in terms of outlier detection; see the next paragraph for a theoretical intuition on this phenomenon. Conversely, a suitable choice of $t$ (e.g., $t = 0.9T$ in the figure) yields a well-performing metric that shares a clear trend aligned with AvgkNN. We found that adopting similar perturbation strength in the linear schedule scenarios (e.g., $t = 0.6T$) yields good performances as well. Below we provide an intuition on why a properly-chosen $t$ is important for the performance of minority score.

**Rationale on the failures of small and too large $t$'s.** Employing weak perturbation (like $t = 0.1T$) leaves little ambiguity in predicting clean samples based on perturbed ones, which enables Tweedie's denoising formula (i.e., Eq. (4)) to offer high-fidelity reconstructions both for majority and minority instances, thereby yielding marginal differences between the their reconstruction losses. On the other hand, when the perturbation is too strong (e.g., $t = T$), it would destroy all information in given clean samples. This makes Tweedie's formula unable to reconstruct information even for majority samples, thus inducing little gap between reconstruction losses of majority and minority samples.

## D.7 ADDITIONAL INVESTIGATIONS ON THE BRAIN MRI DATASET

Figure 13 visualizes real brain MRI images yielding various levels of minority score. Observe the trend that as minority score increases, instances are more likely to contain features associated with lesions of the dataset such as cerebral atrophy, demonstrating the effectiveness of our metric even on this distinctive data modality. Moreover, we see that such brain impairments are often more severe

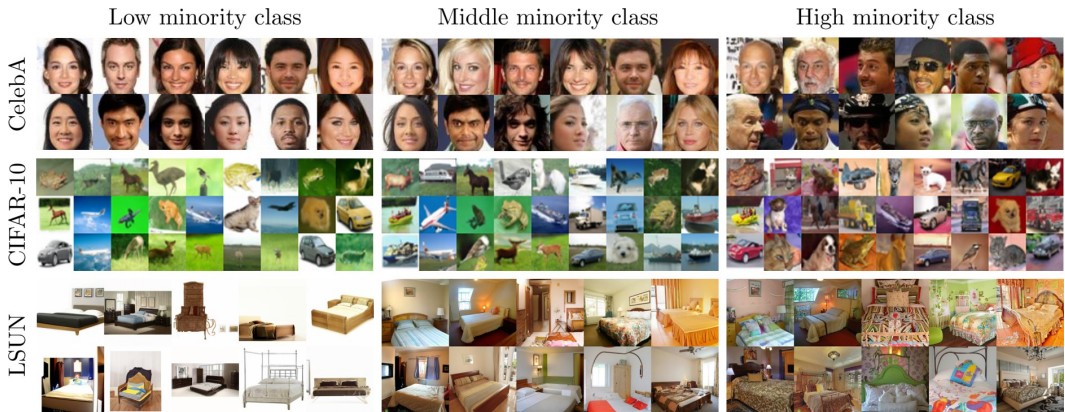

Figure 15: Generated samples from minority guidance over various minority classes. Three different classes are considered: $\tilde{l} = 0$ (left column), $\tilde{l} = 50$ (middle column), and $\tilde{l} = 99$ (right column). We consider three benchmarks: (i) CelebA (top row); (ii) CIFAR-10 (middle row); (iii) LSUN-Bedrooms (bottom row). We share the same random seed for each row.

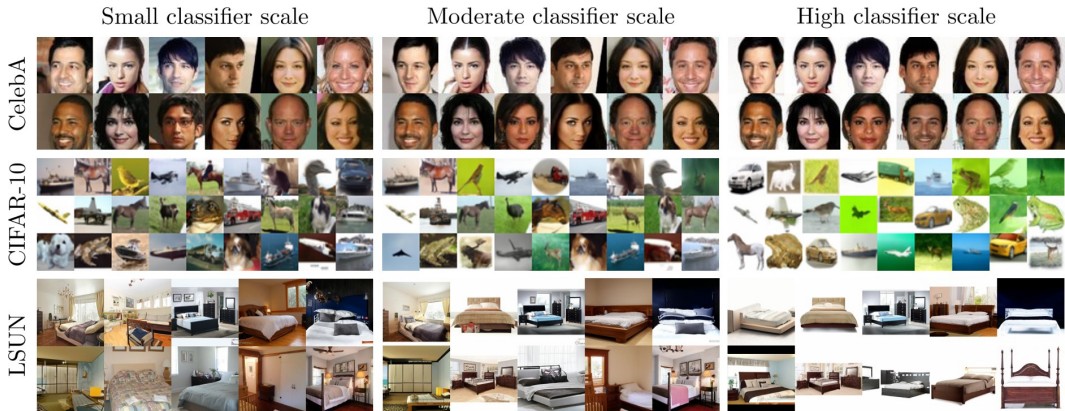

Figure 16: Generated samples by minority guidance over various classifier scales. Three different scales are considered: $w = 0.0$ (left column), $w = 1.0$ (middle column), and $w = 5.0$ (right column). The minority class is fixed to the most common one (i.e., $\tilde{l} = 0$). As in Figure 15, we consider three benchmarks: (i) CelebA (top row); (ii) CIFAR-10 (middle row); (iii) LSUN-Bedrooms (bottom row). We share the same random seed for each row.

in samples with high minority scores, which aligns well with the dataset constitution that rarely contains instances with such significant brain atrophy.

Figure 14 exhibits generated samples by minority guidance considering a range of minority class $\tilde{l}$. Observe that as $\tilde{l}$ increases, the samples tend to have impaired features that correspond to low-density attributes of the data. In the generated samples, we observe the similar correlation as we saw in Figure 13 between the severity of the lesion and minority score, i.e., producing severer impairments with higher minority classes.

We argue that this capability of controlling uniqueness of features is particularly instrumental in medical imaging. In fact, there are numerous cases that demand medical images exhibiting highly unique diseases, such as significant brain atrophy (e.g., the right-side images in Figure 14) which is rarely seen in normal populations. On the other hand, there are instances where we need to generate images with lesions that are not exceedingly rare, like mild cerebral atrophy (e.g., the middle-side images in Figure 14), or even normal images (e.g., the left-side images in Figure 14) without any health issues. Our sampler's ability to control the level of uniqueness may empower us to handle such diverse requirements in medical image generation.

### D.8 CAPABILITY TO LEARN THE GROUND-TRUTH DATA DISTRIBUTION

Since our approach focuses on producing minority instances lying in the tail of a data distribution, one may concern that it could introduce the performance loss against instances within high-density regions. We address this concern herein by investigating the capability of our approach to reproduce all regions of the data manifold not limited to low-density regions. Specifically on CelebA, we evaluate the FID metrics of generated samples by our approach and compare the values with the performances due to the standard sampler (i.e., ancestral sampling). The baseline real data used for the evaluation herein is the entire CelebA dataset (not minority instances used in Table 1.) To encourage generation that covers a wide range of data manifold, we condition our sampler on all ranges of minority classes uniformly at random with a fixed scale $w = 1.0$. See Table 5 for explicit details. Notice in the table that our sampler yields better performance than the baseline DDPM sampler, demonstrating its robustness against the potential performance loss in learning non-minority representations.

| Method | cFID | sFID |
|--------|------|------|
| ADM    | 2.93 | 2.88 |
| Ours   | **2.91** | **2.86** |

Table 5: The capability of reproducing the ground-truth data distribution.

### D.9 INTERACTION WITH OTHER GUIDANCE

One significant benefit of diffusion models lies in the controllable nature of their sampling process, which enables arbitrary conditioning unseen at the training time. Hence, it is important for a guidance technique to preserve such controllability and to harmonize well with other forms of guidance signal. To further evaluate the practical significance of our proposed sampler in this context, we investigate the interactive aspect of our approach with other forms of guidance herein. Specifically, we incorporate an additional (class-conditional) classifier guidance on ImageNet-64 and explore the impact of the supplementary guidance. Table 6 exhibits the accuracy outcomes of an ImageNet classifier applied to generated samples across a spectrum of additional classifier strengths. Notice that as the classifier scale increases, the generated samples increasingly mirror their intended class-conditions, demonstrating the adaptable nature of our approach in accommodating supplementary guidance.

| Scale ($w$) | Accuracy (%) |
|-------------|--------------|
| 0.0 | 46.86 |
| 1.0 | 87.44 |
| 2.0 | 96.26 |
| 4.0 | 99.16 |

Table 6: The impact of an additional classifier guidance.

## E ADDITIONAL EXPERIMENTAL RESULTS

We continue from Section 4.2 of the main paper and provide some additional results. We first exhibit generated samples that visualize the impacts of $\tilde{l}$ and $w$. We then provide the complete results on all five benchmarks where we compare ours with the baselines in a thorough manner.

### E.1 VISUALIZATION OF THE IMPACTS OF $\tilde{l}$ AND $w$

Figure 15 exhibits generated samples by minority guidance considering a variety of the (categorized) minority class $\tilde{l}$. Observe that as $\tilde{l}$ increases, the samples tend to have more rare features that appear similar to the ones observable in the minority samples (e.g., in Figure 2). See the left plot in Figure 3 for a quantitative analysis.

Figure 16 showcases generated samples illustrating the impact of the classifier scale $w$. (see the right plot in Figure 3 for a quantitative analysis). We see that as $w$ increases, the generated samples are more likely to exhibit representative features of the associated minority class (i.e., $\tilde{l} = 0$), such as a frontal-view attribute in CelebA. This observation aligns well with the quantitative result shown in Figure 3 and with the findings presented in Dhariwal & Nichol (2021), which extensively investigates the impact of $w$.

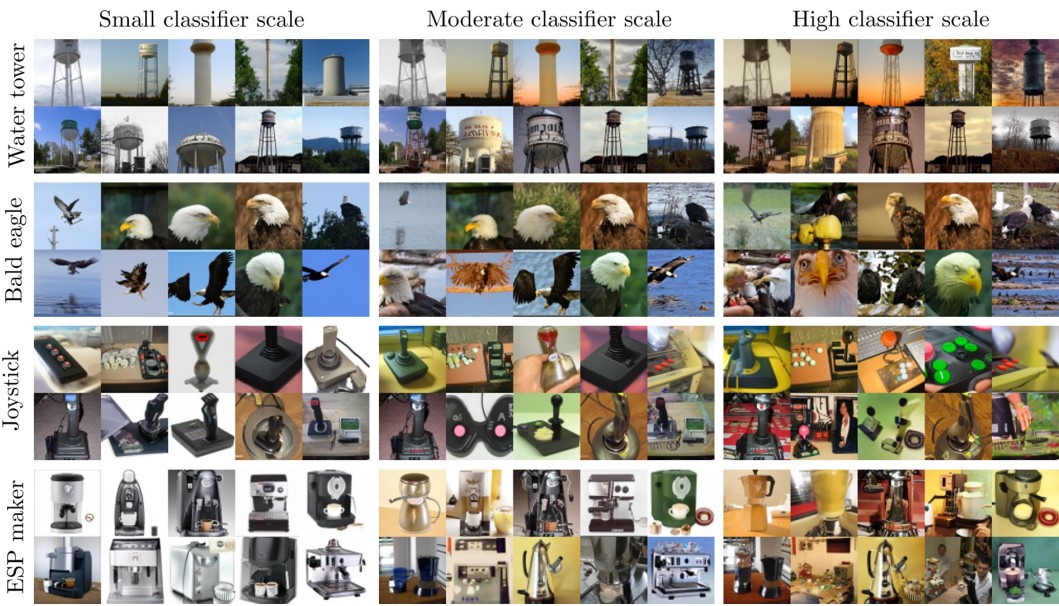

Figure 17: Generated samples by the class-conditional minority guidance over various classifier scales. Three different scales are considered herein: $w = 0.0$ (left column), $w = 1.0$ (middle column), and $w = 2.5$ (right column). The minority class is fixed to the most unique one (i.e., $\tilde{l} = 24$). The same ImageNet classes as in Figure 8 are considered for generation: water tower (top row), bald eagle (top-middle row), joystick (middle-bottom row), and espresso maker (bottom row). We share the same random seed for each row.

Figure 17 further demonstrates the influence of the classifier scale $w$ in the extended minority guidance (i.e., Eq. (17)). It is evident that increasing $w$ leads to a more focused generation of unique features associated with the targeted class $\tilde{l} = 24$, exhibiting a similar trend as in the unconditional setting.

## E.2 COMPARISON WITH THE BASELINES: COMPLETE RESULTS AND ADDITIONAL SAMPLES

Figure 24 exhibits generated samples by our approach and their associated nearest neighboring real samples on CelebA. Notice in the figure that generated features due to our method exhibit a notable level of novelty distinct from those observed in real samples.

Figure 25 visualizes generated samples on CelebA. Observe that the proposed sampler is more likely to generate novel-featured samples that are exhibited in real minority data; see Figures 2 and 9 for instances of the minority features of the dataset. Figure 18 gives a quantitative validation to this observation. We see that minority guidance outperforms the other baselines in terms of generating low-density samples, as demonstrated by yielding more data instances with high values of outlier metrics. The same trend is observed in our CIFAR-10 results; see Figures 26 and 19 for details.

Figure 21 provides a comparison of the performances of neighborhood density measures on the class-conditional ImageNet $64 \times 64$. It is noteworthy that our extended minority guidance (to the class-conditional setting) outperforms the baselines in generating low-density unique samples for all three measures. Notably, our sampler demonstrates superior performance compared to the tailored algorithm by Sehwag et al. (2022), which is specifically designed for the class-conditional setting and not applicable in other contexts like unlabeled-data settings. For completeness, we exhibit the neighborhood density results of LSUN-Bedrooms, which are already reported in the manuscript; see Figure 20 for details.

Figures 22 and 23 exhibit the density measures on the brain MRI dataset. Inspired by Ulyanov et al. (2018) and Naeem et al. (2020) that showcase the power of random embedding spaces for admitting distinctive data modality, we employ randomly-initialized neural networks (e.g., ResNet-50, VGG-

16) for computing density measures for the results in Figure 22. On the other hand, the performance values in Figure 23 are computed in a feature space of DINO (Caron et al., 2021) which is fine-tuned on our brain MRI data (see Section C for details on the fine-tuning setting). Notice that minority guidance outperforms the baselines in generating unique samples for all cases, which corroborates our observations made on Figure 5 (in Section 4.2).

Table 7 provides quality and diversity evaluations on all our considered benchmarks including CIFAR-10. We additionally provide the results on the brain MRI data evaluated based on a fine-tuned feature space of DINO (Caron et al., 2021) on our available brain MRI images. Observe that the same performance benefit of our sampler also applies to the scenarios that are not covered in Table 1, which further strengthen the robustness of our sampler. We found that the performance values of StyleGAN2-ADA (Karras et al., 2020) sometimes diverge on our brain MRI dataset, yielding poor performance values in some metrics (e.g., Precision, cFID on the fine-tuned feature space). This diverging phenomenon is also observed in density distributions of the method, which are widely spreaded over the axes of neighborhood metric values (e.g., as seen in the middle plot in Figures 23). We conjecture that this comes from the inconsistent modalities between natural and medical images, which may hinder proper interpretations of generated MRI features in the ImageNet-pretrained feature space (e.g., hence yielding low precision). Another possible reason is that the fine-tuned feature space (wherein the high-cFID value is measured) may still suffer from the lack of knowledge on the brain MRI modality. While we employed all the available samples (i.e., 13,640 slices) for fine-tuning, it may not be sufficient to construct a well-structured feature space that can admit diverse aspects of the target data, which often requires huge amount of data to achieve.

Table 8 exhibits the evaluation results employing a variety of density metrics for collecting baseline real data. Observe that our sampler yields better (or comparable) performance values under various criteria not limited to AvgkNN, demonstrating the effectiveness of our method in producing instances that are generally unique.

To facilitate a more comprehensive qualitative comparison among the samplers, we provide an extensive showcase of generated samples for all the considered datasets. See Figures 25–29 for details.

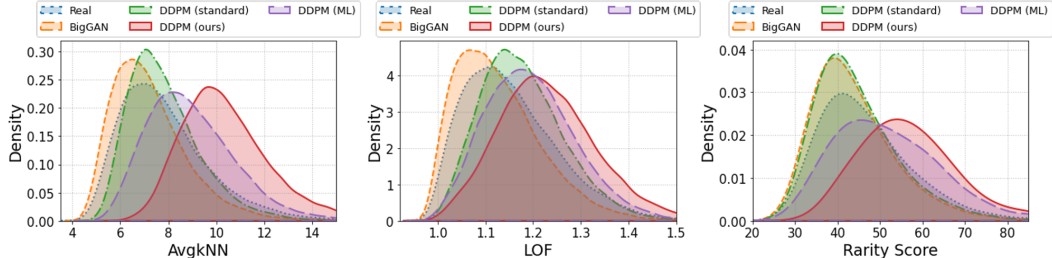

Figure 18: Comparison of neighborhood density on CelebA. "Real" refers to our real-data baseline, the test samples of CelebA. "BigGAN" is our GAN-based generative baseline, BigGAN (Brock et al., 2019). "DDPM (standard)" indicates the diffusion-based generative baseline: ADM (Dhariwal & Nichol, 2021) with the standard sampler (i.e., Eq. (3)) (Ho et al., 2020). "DDPM (ML)" refers to a classifier-guided DDPM sampler conditioning on **M**inority **L**abels. "DDPM (ours)" denotes DDPM with the proposed sampler, minority guidance. "AvgkNN" refers to Average k-Nearest Neighbor, and "LOF" is Local Outlier Factor (Breunig et al., 2000).

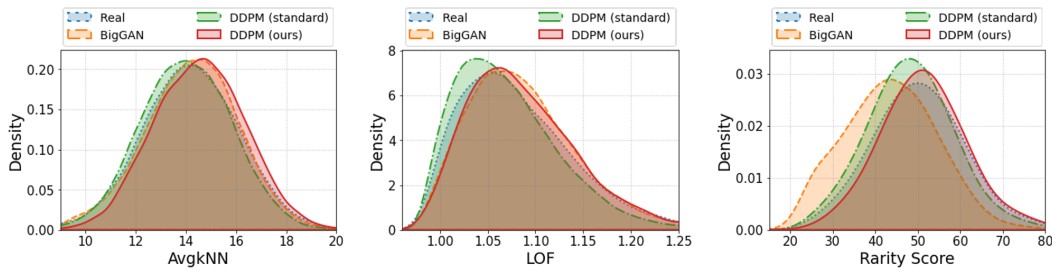

Figure 19: Comparison of neighborhood density on CIFAR-10. "Real" refers to our real-data baseline, the test samples of CIFAR-10. All the remaining settings are the same as those in Figure 18.

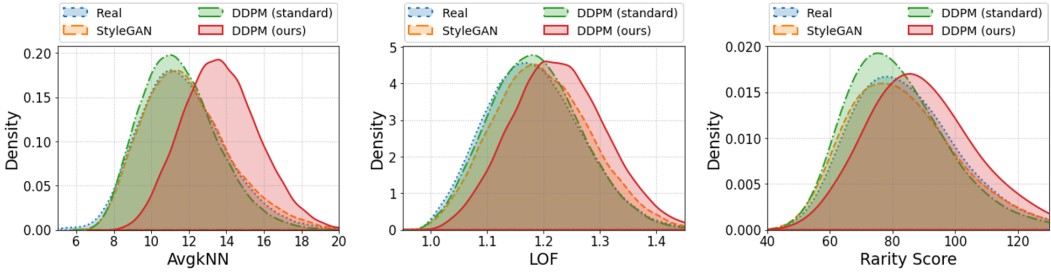

Figure 20: Comparison of neighborhood density on LSUN-Bedrooms. The results are the same as those in Figure 6.

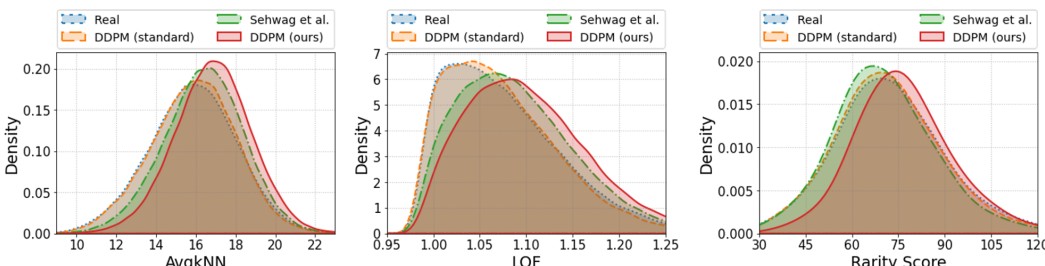

Figure 21: Comparison of neighborhood density on the class-conditional ImageNet. "Real" refers to the validation samples of ImageNet. "Sehwag et al." refers to the low-density sampler proposed in Sehwag et al. (2022). All the remaining settings are the same as those in Figure 18.

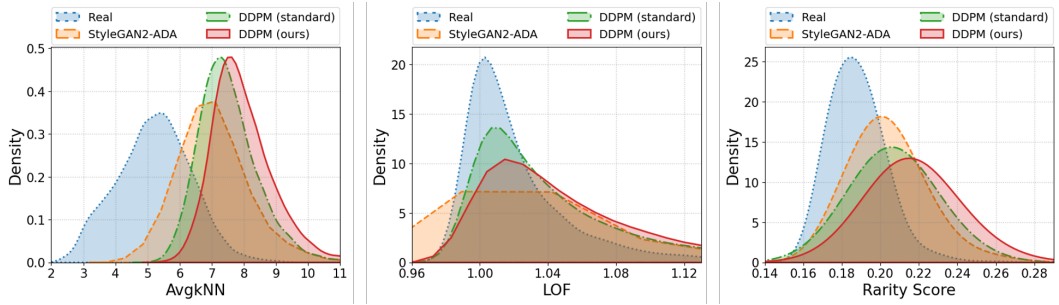

Figure 22: Comparison of neighborhood density on the brain MRI dataset. The density measures are computed in randomly-initialized feature spaces (Ulyanov et al., 2018; Naeem et al., 2020). "Real" refers to the training dataset (i.e., all 13,640 slices). "DDPM (standard)" indicates ADM (Dhariwal & Nichol, 2021) with ancestral sampling. "DDPM (ours)" denotes minority guidance. The higher values, the less likely samples for all three measures.

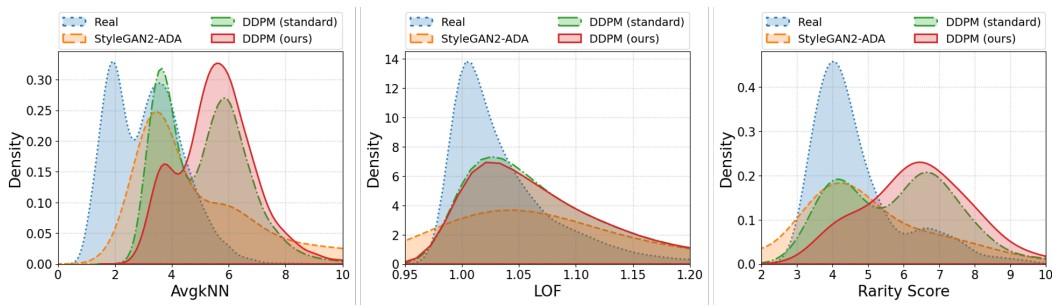

Figure 23: Comparison of neighborhood density on the brain MRI dataset. The density measures exhibited herein are evaluated in a fine-tuned DINO (Caron et al., 2021) feature space on our brain MRI dataset. All the remaining settings are the same as those in Figure 22.

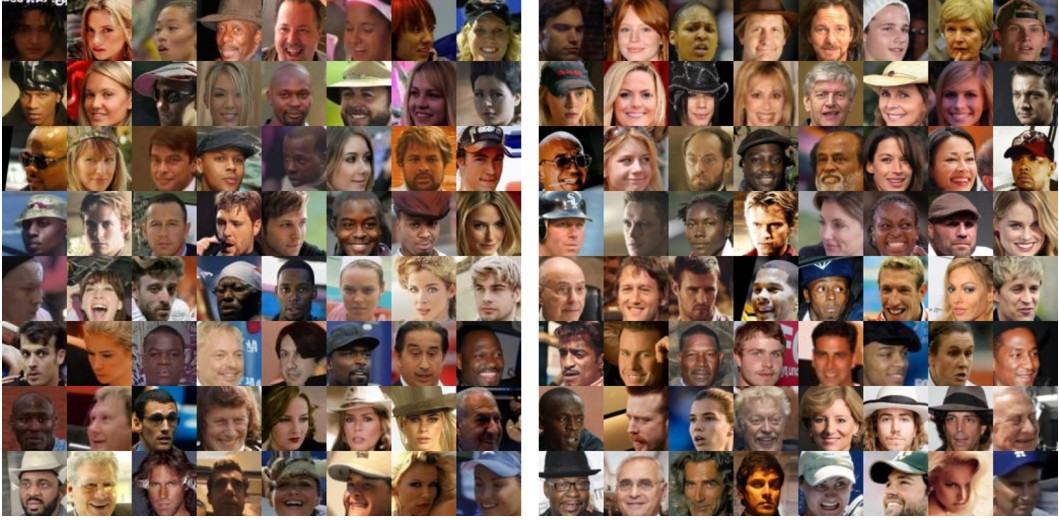

Figure 24: The novelty of generated samples by minority guidance. Generated samples by our approach (left) and their corresponding nearest neighbors in the training dataset of CelebA (right) are visualized.

| Method | cFID | sFID | Prec | Rec | Method | cFID | sFID | Prec | Rec |
|---|---|---|---|---|---|---|---|---|---|
| **CelebA 64×64** | | | | | **ImageNet 64×64** | | | | |
| ADM | 75.05 | 16.73 | **0.97** | 0.23 | ADM | 17.85 | 4.90 | 0.79 | 0.52 |
| ADM-ML | 51.66 | 13.04 | 0.94 | 0.31 | Sehwag et al. | **10.76** | 4.12 | **0.80** | 0.52 |
| BigGAN | 80.41 | 16.51 | **0.97** | 0.19 | Ours | 11.95 | **2.62** | 0.76 | **0.55** |
| Sehwag et al. | 27.97 | 10.17 | 0.82 | **0.42** | | | | | |
| Ours | **26.98** | **8.23** | 0.89 | 0.34 | | | | | |
| **CIFAR-10** | | | | | **ImageNet 256×256** | | | | |
| IDDPM | 31.58 | 10.61 | 0.93 | 0.35 | ADM | 17.41 | 12.85 | **0.87** | 0.35 |
| BigGAN | 34.08 | 12.14 | **0.94** | 0.19 | Sehwag et al. | **13.43** | 9.88 | 0.85 | 0.42 |
| Ours | **29.95** | **9.63** | 0.92 | **0.41** | Ours | 13.83 | **7.82** | 0.85 | **0.45** |
| **CIFAR-10-LT** | | | | | **Brain MRI 256×256** | | | | |
| DDPM | 75.71 | 44.26 | **0.95** | 0.23 | ADM | 21.71 | 14.30 | 0.82 | 0.37 |
| Qin et al. | 64.93 | **41.41** | 0.92 | **0.36** | StyleGAN2a | 22.69 | 23.07 | 0.53 | 0.28 |
| Ours | **63.34** | 42.53 | **0.95** | 0.24 | Ours | **19.73** | **13.36** | **0.84** | **0.47** |
| **LSUN Bedrooms 256×256** | | | | | **Brain MRI 256×256 (FT f-space)** | | | | |
| ADM | 62.83 | 7.49 | 0.89 | **0.16** | ADM | 20.64 | – | **0.87** | 0.77 |
| LDM | 63.61 | 7.19 | **0.90** | 0.13 | StyleGAN2a | 315.19 | – | 0.69 | 0.61 |
| StyleGAN | 56.45 | **5.96** | 0.87 | 0.13 | Ours | **11.83** | – | 0.86 | **0.79** |
| Ours | **41.52** | 6.67 | 0.87 | 0.10 | | | | | |

Table 7: Comparison of sample quality and diversity for generating minority samples. The results on all considered benchmarks are exhibited herein. "ADM-ML" refers to a classifier-guided ancestral sampling conditioning on **M**inority **L**abels. "StyleGAN2a" indicates StyleGAN2-ADA (Karras et al., 2020). "FT f-space" refers to the evaluations made in an MRI-adapted feature space constructed by fine-tuning the pretrained DINO (Caron et al., 2021) on our brain MRI benchmark. The best results are marked in bold.

| Method | cFID | sFID | Prec | Rec | Method | cFID | sFID | Prec | Rec |
|---|---|---|---|---|---|---|---|---|---|
| **CelebA 64×64 (LOF)** | | | | | **LSUN Bedrooms 256×256 (MS)** | | | | |
| ADM | 32.50 | 10.21 | 0.94 | 0.30 | ADM | 28.20 | 11.74 | 0.54 | 0.33 |
| ADM-ML | 30.37 | 9.76 | 0.88 | 0.39 | StyleGAN | 24.19 | 8.43 | 0.57 | 0.33 |
| BigGAN | 35.05 | 10.16 | **0.95** | 0.29 | Ours | **6.92** | **5.53** | **0.70** | **0.36** |
| Ours | **25.41** | **8.90** | 0.80 | **0.43** | | | | | |
| **CelebA 64×64 (RS)** | | | | | **ImageNet 64×64 (LOF)** | | | | |
| ADM | 60.68 | 12.75 | **0.98** | 0.22 | ADM | 11.55 | 4.05 | **0.83** | 0.52 |
| ADM-ML | 32.45 | 8.16 | 0.97 | 0.31 | Sehwag et al. | **8.28** | 3.45 | **0.83** | 0.52 |
| BigGAN | 67.07 | 13.12 | **0.98** | 0.18 | Ours | 8.96 | **2.42** | 0.80 | **0.55** |
| Ours | **20.68** | **7.71** | 0.93 | **0.32** | | | | | |
| **CelebA 64×64 (MS)** | | | | | **ImageNet 64×64 (MS)** | | | | |
| ADM | 58.15 | 16.37 | **0.91** | 0.34 | ADM | 14.67 | 6.09 | 0.70 | 0.62 |
| ADM-ML | 30.32 | 10.99 | 0.87 | 0.43 | Sehwag et al. | 12.57 | 5.30 | 0.70 | 0.62 |
| BigGAN | 63.16 | 16.15 | **0.91** | 0.29 | Ours | **5.88** | **2.34** | **0.71** | **0.69** |
| Ours | **8.88** | **5.37** | 0.85 | **0.49** | | | | | |
| **CIFAR-10 (LOF)** | | | | | **Brain MRI 256×256 (MS)** | | | | |
| IDDPM | 17.31 | 8.56 | **0.93** | 0.37 | ADM | 24.02 | 21.10 | 0.78 | 0.35 |
| BigGAN | 31.05 | 10.74 | **0.93** | 0.22 | StyleGAN2a | 26.36 | 19.82 | 0.47 | 0.28 |
| Ours | **17.05** | **8.43** | 0.91 | **0.42** | Ours | **14.50** | **16.99** | **0.84** | **0.40** |
| **CIFAR-10 (MS)** | | | | | **Brain MRI 256×256 (MS; FT f-space)** | | | | |
| IDDPM | 25.62 | 13.53 | **0.83** | 0.51 | ADM | **30.73** | – | 0.81 | **0.79** |
| BigGAN | 46.13 | 15.93 | 0.82 | 0.31 | StyleGAN2a | 281.08 | – | 0.63 | 0.67 |
| Ours | **15.16** | **8.86** | **0.83** | **0.59** | Ours | 35.49 | – | **0.84** | **0.79** |

Table 8: Comparison of sample quality and diversity for generating minority samples. Various different criteria (other than AvgkNN which is covered in Tables 1 and 7) are used for constructing real baseline minority data. "LOF" indicates the results derived with the real baseline data that yields the highest LOF values. Similarly, "RS" and "MS" are the evaluations based on the baseline data inducing the highest Rarity Score and minority score values, respectively. "ADM-ML" refers to a classifier-guided ancestral sampler conditioning on **M**inority **L**abels. "StyleGAN2a" indicates StyleGAN2-ADA (Karras et al., 2020). "FT f-space" refers to the evaluations made in an MRI-adapted feature space constructed by fine-tuning the pretrained DINO (Caron et al., 2021) on our brain MRI benchmark. The best results are marked in bold.

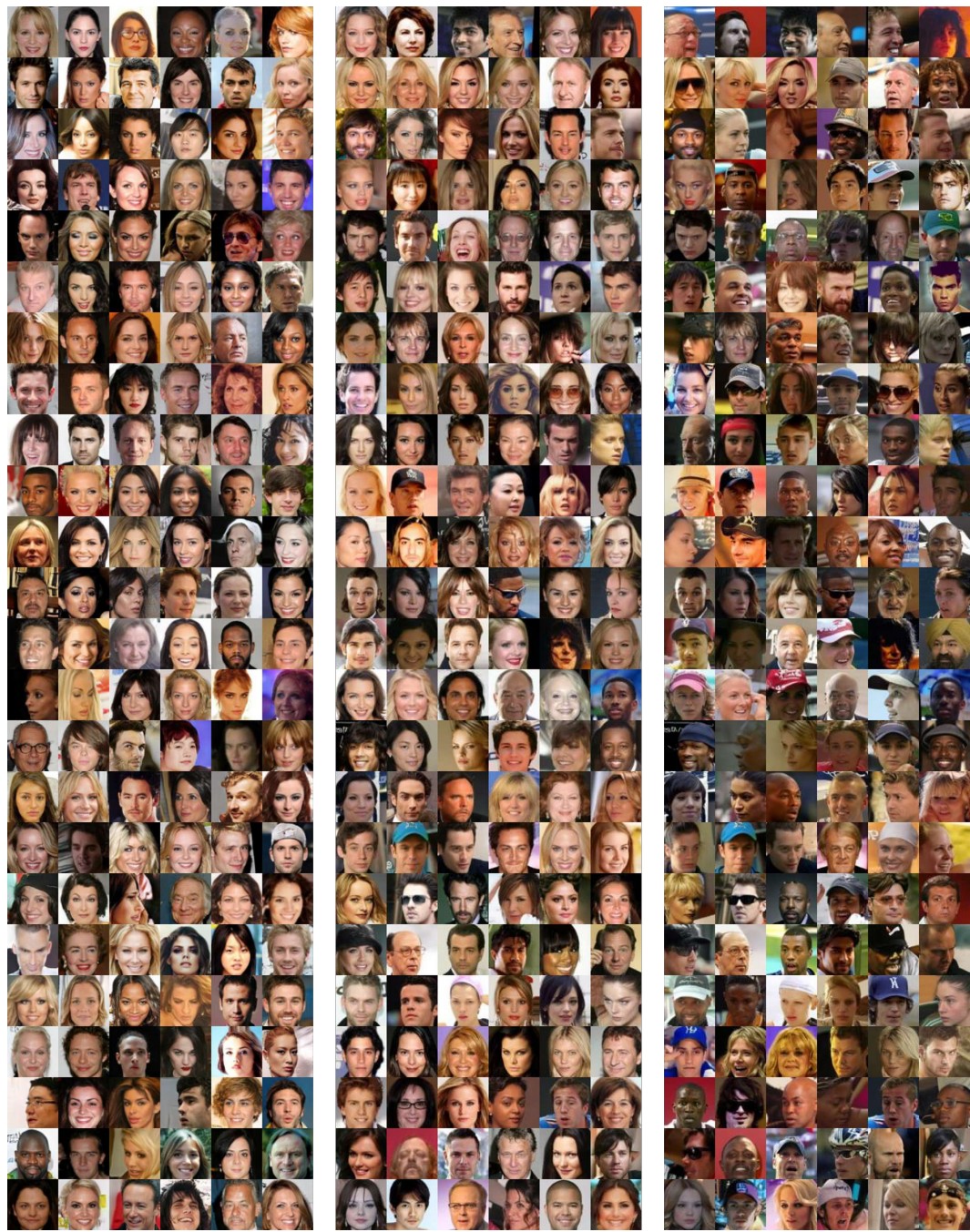

Figure 25: Sample comparison on CelebA. Generated samples by BigGAN (Brock et al., 2019) (left), ADM (Dhariwal & Nichol, 2021) with ancestral sampling (middle), and minority guidance (right) are exhibited. We share the same random noise for the generations based on the DDPM-based samplers (i.e., middle and right).

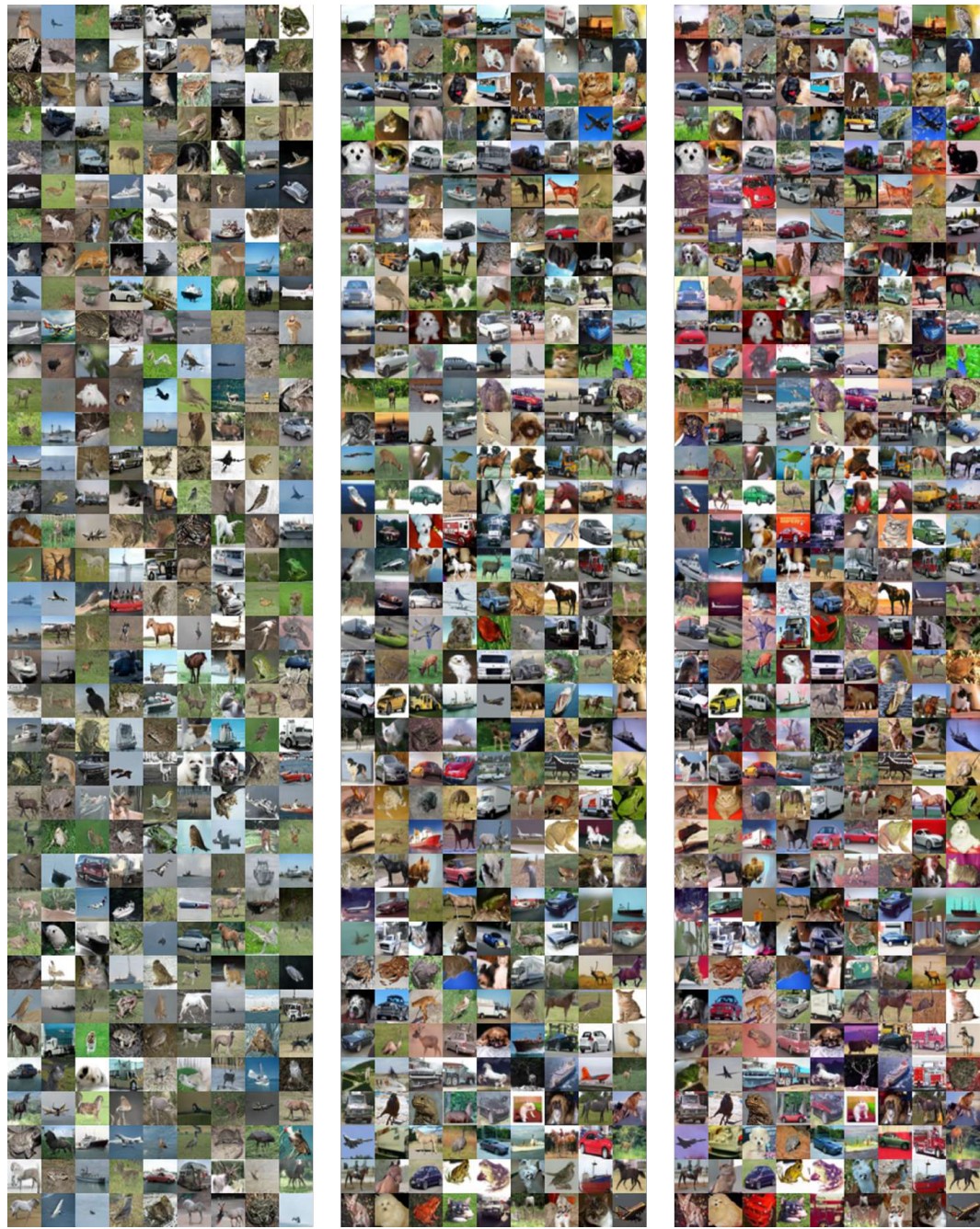

Figure 26: Sample comparison on CIFAR-10. Generated samples by BigGAN (Brock et al., 2019) (left), IDDPM (Nichol & Dhariwal, 2021) with ancestral sampling (middle), the proposed sampler (right) are visualized herein. We share the same random noise for the generations based on the DDPM-based samplers (i.e., middle and right).

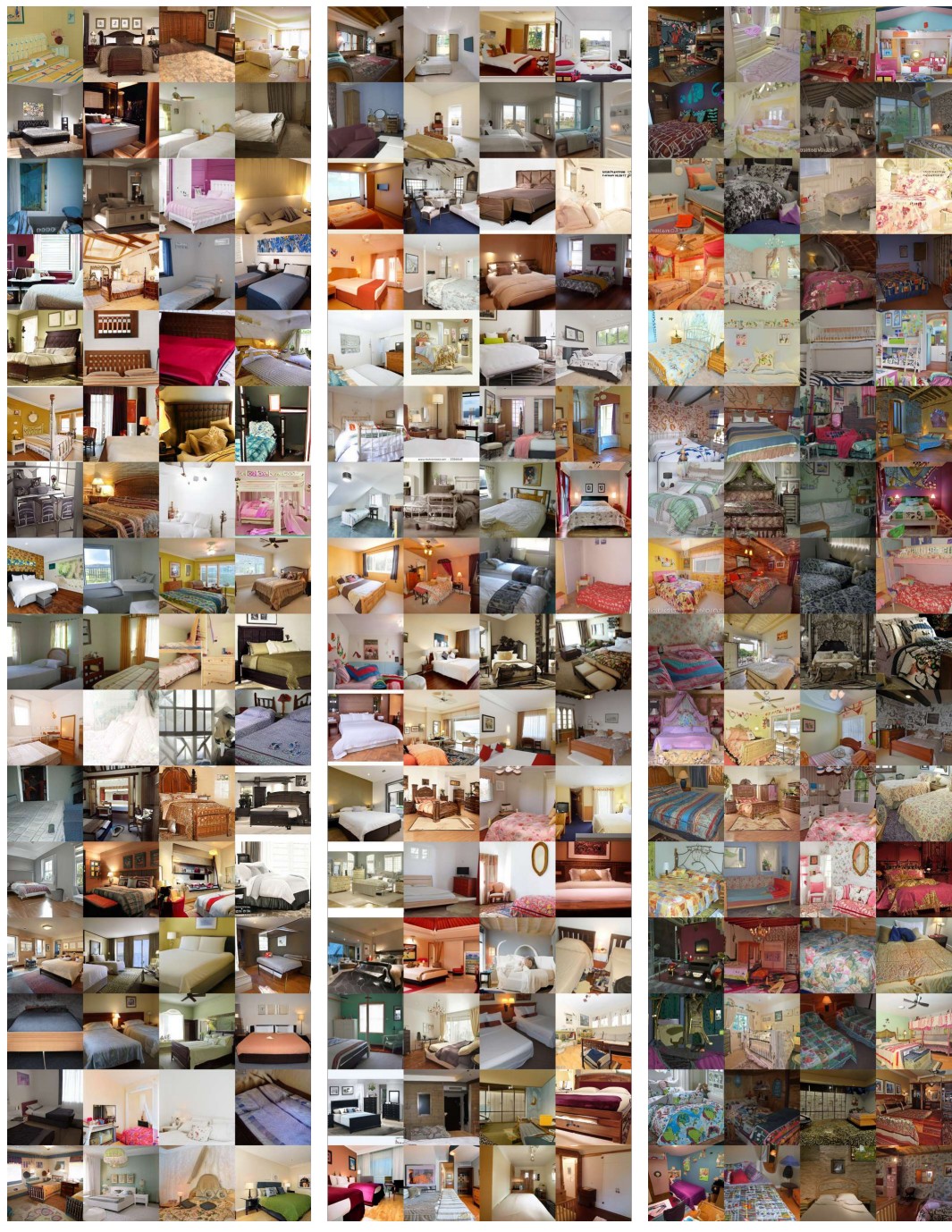

Figure 27: Additional images for comparison on LSUN-Bedrooms. Generated samples by Style-GAN (Karras et al., 2019) (left), ADM (Dhariwal & Nichol, 2021) with ancestral sampling (middle), and minority guidance (right) are exhibited. We use the same random seed for the diffusion-based samplers (i.e., middle and right).

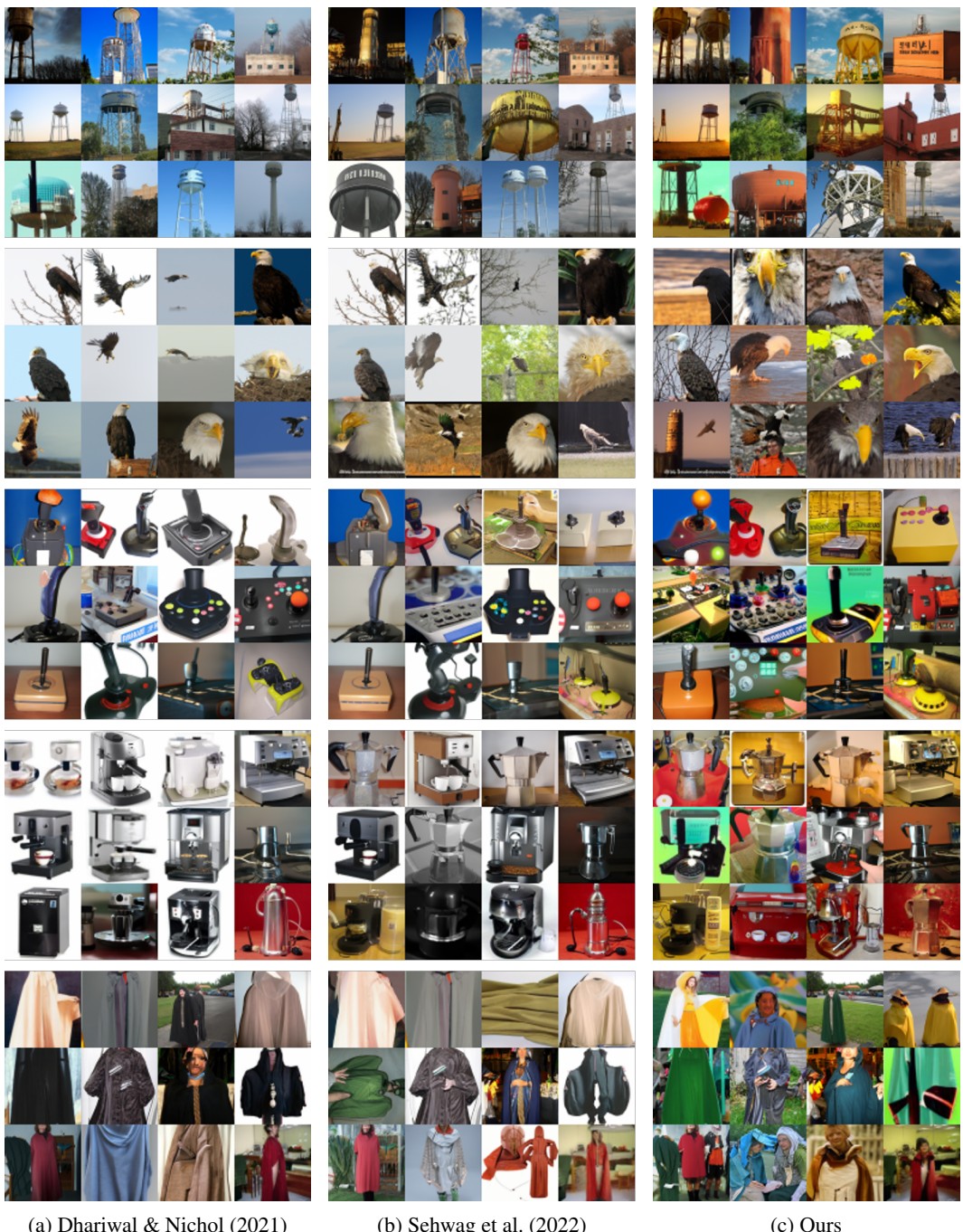

(a) Dhariwal & Nichol (2021)        (b) Sehwag et al. (2022)        (c) Ours

Figure 28: Sample comparison on the class-conditional ImageNet. Generated samples from five classes are exhibited: (i) Water tower (top row); (ii) Bald eagle (top-middle row); (iii) Joystick (middle row); (iv) Espresso maker (middle-bottom row); (v) Cloak (bottom row). For each row, we share the same random seed across all three methods.

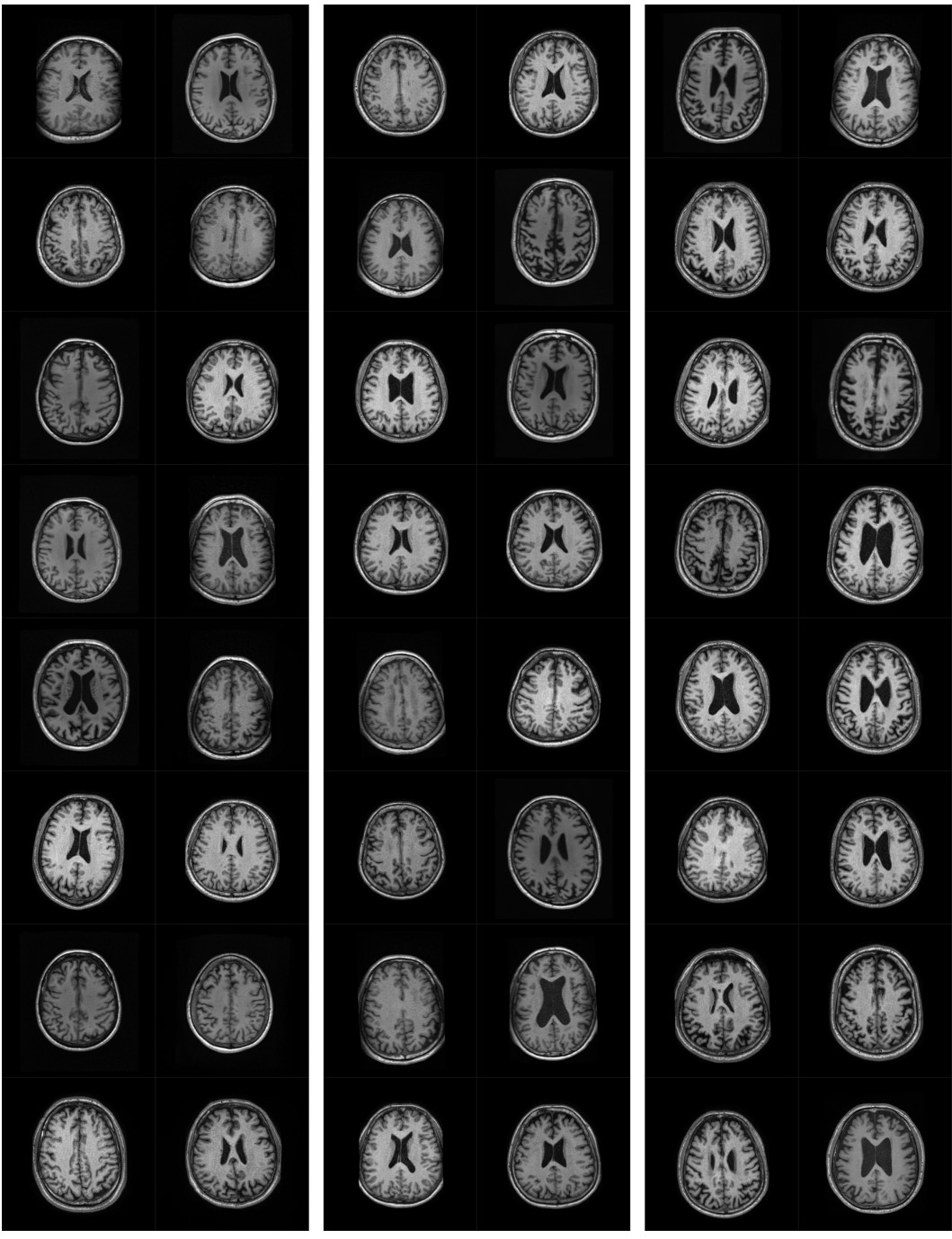

Figure 29: Additional images for comparison on the brain MRI dataset. Generated samples by StyleGAN2-ADA (Karras et al., 2020) (left), ADM (Dhariwal & Nichol, 2021) with ancestral sampling (middle), and minority guidance (right) are exhibited. We use the same random seeds for the diffusion-based samplers (i.e., middle and right).

