# OpenReview forum: "Don't Play Favorites: Minority Guidance for Diffusion Models"
_ICLR.cc/2024/Conference — ICLR 2024 poster_

### Official Review · Reviewer_xCB3 · 2023-10-29

**Soundness:** 3 good
**Presentation:** 3 good
**Contribution:** 3 good
**Rating:** 6
**Confidence:** 4

**Summary:**

This article primarily aims to address the issue of diffusion models' inability to generate data samples in low-density regions of the data manifold. Initially, they illustrate that Tweedie's denoising formula tends to generate the majority of samples, which lie in high-density regions of the manifold. Building upon this, they devise the minority score to quantify the uniqueness of the features of a given data instance. Drawing inspiration from classifier guidance, they incorporate minority guidance into the diffusion model to generate samples in low-density regions.

**Strengths:**

1. The paper is easy to read and generally well-written.
2. The paper evaluates the proposed method on benchmark real datasets and demonstrates the effectiveness of their approach in demanding real-world scenarios, such as medical imaging.

**Weaknesses:**

1. Given the scarcity of real samples in low-density regions, how is it ensured that the proposed method doesn't suffer from overfitting? In other words, how can the authenticity and diversity of the generated samples in low-density regions be guaranteed?
2. The authors should provide more experimental details. For instance, what is the size of the subset of real data used for validation? In the experimental section,what levels of minority score were chosen for generating the data?
3. In Fig. 1(a), what perturbation timestep was used to generate the noise samples displayed in the middle, 0.9T or T? The choice of 't' is crucial in Section 3.1; the authors should consider showing specific images of $x_t$, $x_0$ and $\hat x_0$ or provide theoretical evidence that  is indeed the reconstructed image.
4. In the experimental section, the method proposed in this paper did not achieve satisfactory results in terms of both prec and rec. The authors should provide further explanations.
5. The nomenclature in this paper should be more standardized. For instance, in Section 4.2, there is an error in the representation of $\tilde l$.

**Questions:**

Please refer to Weaknesses.

---

> ### Author Response · Authors · 2023-11-18
>
> We would like to express our sincere gratitude for your insightful and detailed feedback which indeed helped us to improve the manuscript. Below we provide point-by-point responses.
>
> ---
>
> >**W1.** Given the scarcity of real samples in low-density regions, how is it ensured that the proposed method doesn't suffer from overfitting? In other words, how can the authenticity and diversity of the generated samples in low-density regions be guaranteed?
>
> To address your concern, we have conducted an investigation into the nearest neighboring real samples of our generated instances. The results indicate that samples from our method exhibit a notable level of novelty. See Figure 24 in our revised manuscript for instance. Furthermore, our approach demonstrates favorable diversity performance as confirmed by our quantitative results (e.g., in Table 1), providing additional support for the robustness of our methodology.
>
> ---
>
> >**W2.** The authors should provide more experimental details. For instance, what is the size of the subset of real data used for validation? In the experimental section,what levels of minority score were chosen for generating the data?
>
> The reviewer is kindly reminded that the suggested information are already contained in Section C. Specifically, the sizes of minority real data used for evaluation are chosen based on the volume of the entire datasets. For instance, we employ the most unique (yielding the highest AvgkNN) 50K real samples on huge benchmarks like LSUN-Bedrooms, while using the rarest 10K (or 5K) samples for relatively small datasets like CelebA. See Section C for more details. The minority classes are often randomly selected from a pool of highly unique classes, e.g., the most unique 10 classes. See Table 2 in Section C for explicit details.
>
> ---
>
> >**W3.** In Fig. 1(a), what perturbation timestep was used to generate the noise samples displayed in the middle, 0.9T or T? The choice of 't' is crucial in Section 3.1; the authors should consider showing specific images of xt, x0 and x0hat or provide theoretical evidence that is indeed the reconstructed image.
>
> $0.9T$ was used for the images in Figure 1.(a). The images in “Original” column in Figure 1.(a) correspond to $\boldsymbol{x}_0$, and the ones in “Perturbed” column are $\boldsymbol{x}_t$ (perturbed with $t=0.9T$). The images in “Reconstructed” column visualizes $\hat{\boldsymbol{x}}_0$ that are reconstructed from $\boldsymbol{x}_t$ in “Perturbed” column. We have reflected this in our revised manuscript; see Figure 1 therein.
>
> ---
>
> >**W4.** In the experimental section, the method proposed in this paper did not achieve satisfactory results in terms of both prec and rec. The authors should provide further explanations.
>
> Despite inferiorities in some metrics (like precision), our method works better in the holistic viewpoint as exhibited in the significant gain in comprehensive metrics (e.g., cFID), attaining better trade-offs between quality and diversity of low-density samples. Also, we would like to note that the bad recall performance in LSUN-Bedrooms is mainly because we focused on a narrow range of minorities (i.e., $\tilde{l} = 99$) for generation (see Table 2 in Section C for the setting). This could be readily resolved by considering a wide range of minority classes (e.g., $\tilde{l} \sim \text{unif} \lbrace 90, 99 \rbrace$).
>
> ---
>
> >**W5.** The nomenclature in this paper should be more standardized. For instance, in Section 4.2, there is an error in the representation of \tilde{l}.
>
> Fixed. Thanks for pointing out.

---

> > ### Comment · Reviewer_xCB3 · 2023-11-22
> > **Response to authors**
> >
> > The authors have addressed my concerns, thus, I keep my rating.

---

> > > ### Author Response · Authors · 2023-11-22
> > > **Dear Reviewer xCB3**
> > >
> > > Thank you for your valuable feedback and giving us a positive score. We are pleased that our response has addressed your concerns.

---

### Official Review · Reviewer_Xf5u · 2023-11-01

**Soundness:** 3 good
**Presentation:** 3 good
**Contribution:** 3 good
**Rating:** 6
**Confidence:** 4

**Summary:**

In this work the authors proposed a guidance to diffusion models so it is encouraged to generate the minority data. In doing so, the author opposed the use of the distance between original data and the denoised version of the same data by a diffusion model as a minority score, inspired by Tweedie’s denosing formulae. Data with larger such score is used to train a auxiliary classifier, which is used to guide the generation with commonly used classifier-guidance approach. Substantial theoretical and empirical results are shown to demonstrate the superiority, especially quantitative, of the proposed method.

**Strengths:**

Overall, this paper is okay to read. It is a neat combination of straightforward ideas that produces good results: identifying uniqueness with distance between clean and denoised data, marking the data points that are showing minority data, and training a classifier of minority data to guide the generation.

Particularly, the problem of generating samples belonging to the minority part of the data domain is an interesting one with potential applications. Components in the proposed work are existing ideas (Tweedie’s formula inspired deonsing, training auxiliary classifier to guide the inference process), but the quantitative quality are consistent and should give readers some insights. Furthermore I’d like to praise the authors for detailed explanation of the method, theories behind and extra examples in appendix.

However there are some issues (see weaknesses and questions below) in the manuscript. Overall my assessment of this manuscript is borderline, but if my concerns are addressed I’m happy to adjust accordingly.

**Weaknesses:**

Overall the technical contributions are somewhat limited, as ideas are from existing works. This may not be a issue on its own as long as the work presents an interesting approach in the whole, but a few missing details hurts, as follows:

It is unclear how the minority classifier guidance plays with other classifier guidance or classifier-free guidance. From an application perspective, it’s important that users can control the generation through guidance, and a mere generation of minority data, while still being useful for say data augmentation, may limit its value for the end-user if the user cannot control the generation effectively. However, now only in Appendix E some examples are shown for conditional generation and details/quantitative evaluates are omitted. This makes it hard to assess the broader impact of this work.

Also, it would be better to discuss the rationale behind minority score — the exact motivation of having a minority data generation remains not elaborated.

**Questions:**

See weakness

---

> ### Author Response · Authors · 2023-11-18
>
> We would like to thank your very thoughtful and detailed comments, which helped us to improve the manuscript. Below we provide point-by-point responses.
>
> ---
>
> >**W1.** It is unclear how the minority classifier guidance plays with other classifier guidance or classifier-free guidance. From an application perspective, it’s important that users can control the generation through guidance, and a mere generation of minority data, while still being useful for say data augmentation, may limit its value for the end-user if the user cannot control the generation effectively. However, now only in Appendix E some examples are shown for conditional generation and details/quantitative evaluates are omitted. This makes it hard to assess the broader impact of this work.
>
> In response to your insightful suggestion, we now provide new results that exhibit the interactive aspect of our approach with other forms of guidance. Specifically we incorporated an additional classifier guidance on ImageNet-64 and found that the impact of classifier guidance persists even in the presence of our minority guidance. Below we exhibit the accuracy outcomes of an ImageNet classifier applied to generated samples across a spectrum of additional classifier strengths:
>
> | CG scale       |     Class accuracy  |
> | ------------------ | ------- |
> | w = 0.0            | 46.86 \%  |
> | w = 1.0            | 87.44 \% |
> | w = 2.0            | 96.26 \% |
> | w = 4.0            | 99.16 \% |
>
> Notice that as the classifier scale increases, the generated samples increasingly mirror their intended class-conditions, demonstrating the adaptable nature of our approach in accommodating supplementary guidance. We now included this point in our revision (Section D.9, highlighted in blue).
>
> ---
>
> >**W2.** Also, it would be better to discuss the rationale behind minority score — the exact motivation of having a minority data generation remains not elaborated.
>
> As you may guess, the ability to generate minority data holds considerable importance in diverse real-world applications. For instance, it can enhance the predictive capabilities of medical diagnosis models by offering additional instances of rare conditions. Another noteworthy use-case pertains to fairness, as minority instances often align with social vulnerabilities. Augmenting their representations in the data can contribute to promoting fairness in various downstream applications. Moreover, unique features contained within low-likelihood instances are important in use-cases like creative AI applications [Ref-1], where the ability to generate samples with exceptional creativity is an essential component of the task. We now incorporated these points in our revised manuscript (the 1st paragraph in Section 1, highlighted in blue).
>
> ---
>
> **References**
>
> [Ref-1] Rarity score: A new metric to evaluate the uncommonness of synthesized images, ICLR 2023.

---

### Official Review · Reviewer_F8nE · 2023-11-07

**Soundness:** 3 good
**Presentation:** 2 fair
**Contribution:** 2 fair
**Rating:** 6
**Confidence:** 4

**Summary:**

This work tackles the problem of minority data generation using diffusion models. First, they propose a metric that defines the uniqueness of a sample. Then, they propose a classifier guidance approach based on the proposed metric to generate minority samples. The proposed metric and guidance are based on Tweedie's formula for denoising. The method is evaluated on 4 datasets for the minority generation task.

**Strengths:**

1. The theoretical background is well explained and supports the claims.

2. The paper is well-written and easy to follow.

3. The idea of using Tweedie's denoising method for minority sample generation seems to be novel to the best of my knowledge.

**Weaknesses:**

1. Evaluation: The baselines are limited to a few outdated works (StyleGAN 2020, DDPM 2020). The only recent baseline is by Sehwag 2022 on ImageNet which outperforms the proposed method in two out of four metrics. All methods in Tab. 1 are evaluated for the minority generation task, which despite being old, outperform the proposed method in some metrics. More recent methods, such as LDM [a], which is not even cited, can generate minority samples to some extent.

2. Literature Review: The literature review is limited. There are many related methods that are not cited, contrasted against, or compared to:

[a] Rombach, Robin, et al. "High-resolution image synthesis with latent diffusion models." CVPR 2022.

[b] Huang, Gaofeng, and Amir Hossein Jafari. "Enhanced balancing GAN: Minority-class image generation." Neural computing and applications 35.7 (2023): 5145-5154.

[c] Samuel, Dvir, et al. "Generating Images Of Rare Concepts Using Pretrained Diffusion Models. arXiv 2023.

**Questions:**

1. The method is evaluated quantitatively for the minority image generation task. Doesn't it lead to loss of performance for generating long-tail data?

2. Were the minority samples only chosen based on the minority score? Did the authors perform any verification on whether they are all actually minorities? There are some samples shown for different minority scores, but I was wondering if there was a more systematic verification process or not.

---

> ### Author Response · Authors · 2023-11-18
> **Official Comment by Authors (1/2)**
>
> We would like to express our sincere gratitude for providing useful comments, which indeed helped us to improve the manuscript. Below we provide point-by-point responses.
>
> ---
>
> >**W1-1.** Evaluation: The baselines are limited to a few outdated works (StyleGAN 2020, DDPM 2020).
>
> The reviewer is kindly reminded that as mentioned in Section C, the frameworks that we used for our diffusion baselines are actually ADM [Ref-1] (except for CIFAR-10 where IDDPM [Ref-2] was employed) which exhibit comparable performance to popular frameworks like LDM [Ref-3]. The reason we call such baselines as “DDPM” is to indicate that they employ ancestral sampler (which is often called “the DDPM sampling” [Ref-4]) in the latent space. We now modified our manuscript to clarify this point (Section 4, highlighted in blue).
>
> ---
>
> >**W1-2.** The only recent baseline is by Sehwag 2022 on ImageNet which outperforms the proposed method in two out of four metrics.
>
> While the methodology presented by Sehwag et al. (2022) demonstrates a performance comparable to ours on ImageNet-64, a more expansive evaluation across diverse benchmarks reveals the overall superiority of our proposed approach. Specifically we found that our sampler yields better results than Sehwag et al. (2022) on CelebA; see the table below for details:
>
> | Method                           | cFID   | sFID   | Prec  | Rec   |
> | -------------------------------- | ------ | ------ | ----- | ----- |
> | BigGAN               | 80.41  | 16.51   | **0.97**  | 0.19  |
> | ADM [Ref-1]         | 75.05  | 16.73   | **0.97**   | 0.23  |
> | ADM-ML                | 51.66  | 13.04  | 0.94   | 0.31  |
> | Sehwag et al. (2022)              | 27.97  | 10.17  | 0.82  | **0.42**  |
> | Ours                              | **26.98**  | **8.23**  | 0.89  | 0.34  |
>
> Furthermore, our shortcomings in specific metrics (as exhibited in the results on ImageNet-64) are notably mitigated in the context of ImageNet-256, a more challenging (and often practically significant) benchmark than ImageNet-64. Below we exhibit the table for the comparison on ImageNet-256:
>
> | Method                           | cFID   | sFID   | Prec  | Rec   |
> | -------------------------------- | ------ | ------ | ----- | ----- |
> | ADM [Ref-1]                       | 17.41  | 12.85  | **0.87**  | 0.35  |
> | Sehwag et al. (2022)              | **13.43**  | 9.88  | 0.85 | 0.42  |
> | Ours                                  | 13.83  | **7.82**  | 0.85  | **0.45** |
>
> ---
>
> >**W1-3.** All methods in Tab. 1 are evaluated for the minority generation task, which despite being old, outperform the proposed method in some metrics.
>
> We would like to assure the reviewer that despite inferiorities in some metrics (like precision), our method works better in the holistic viewpoint as exhibited in the significant gain in comprehensive metrics (e.g., cFID), attaining better trade-offs between quality and diversity of low-density samples.
>
> ---
>
> >**W1-4.** More recent methods, such as LDM [a], which is not even cited, can generate minority samples to some extent.
>
> The reviewer is kindly reminded that LDM would exhibit similar performance to our baseline diffusion frameworks (e.g., ADM [Ref-1]) in generating minority instances, since the generation of LDM is also based on ancestral sampling (like ADM) that predominantly samples from high-density regions of data manifold (as claimed in [Ref-4]). To support this, we now conduct new experiments to compare ours with LDM; see the table below for instance:
>
> | Method                           | cFID   | sFID   | Prec  | Rec   |
> | -------------------------------- | ------ | ------ | ----- | ----- |
> | ADM [Ref-1]                    | 62.83 | 7.49 | 0.89  | 0.16  |
> | LDM [Ref-3]                     | 63.61  | 7.19  | **0.90** | 0.13  |
> | StyleGAN                          | 56.45  | **5.96**  | 0.87  | 0.13  |
> | Ours                                  | **41.52**  | 6.67  | 0.87  | **0.10**  |
>
> We observe the same trend of our benefit, yielding significantly better holistic performance metrics compared to LDM. We now included these results in our revision (Table 1 in Section 4.2, highlighted in blue).
>
> ---
>
> >**W2.** Literature Review: The literature review is limited. There are many related methods that are not cited, contrasted against, or compared to:
> [a] Rombach, Robin, et al. "High-resolution image synthesis with latent diffusion models." CVPR 2022.
> [b] Huang, Gaofeng, and Amir Hossein Jafari. "Enhanced balancing GAN: Minority-class image generation." Neural computing and applications 35.7 (2023): 5145-5154.
> [c] Samuel, Dvir, et al. "Generating Images Of Rare Concepts Using Pretrained Diffusion Models. arXiv 2023.
>
> Thanks for pointing out. We now included comparisons with the pointed approaches in our revised manuscript. See related work therein for details (highlighted in blue).

---

> ### Author Response · Authors · 2023-11-18
> **Official Comment by Authors (2/2)**
>
> >**Q1.** The method is evaluated quantitatively for the minority image generation task. Doesn't it lead to loss of performance for generating long-tail data?
>
> As you may guess, our quantitative evaluation is specifically tailored to assess the capability of producing minority instances, aligning with the primary objective of our study. Therefore, one may expect some level of performance loss in other high-density region. Nonetheless, the suspected performance loss was not observed in our sampler. Specifically when conditioned on all ranges of minority classes (uniformly at random), we found that the proposed sampler yields instances that are close to the ground truth data distribution compared to the standard diffusion sampler. See the table below for instance where we exhibit the performance values evaluated using the entire CelebA:
>
> | Method                           | cFID   | sFID   |
> | -------------------------------- | ------ | ------ |
> | ADM [Ref-1]                          | 2.93 | 2.88  |
> | Ours                                  | **2.91**  | **2.86**  |
>
> We see that our sampler yields better performance than the baseline DDPM sampler (on the ADM backbone), demonstrating its robustness against the potential performance loss in learning non-minority representations. We now included this point in our revision (Section D.8, highlighted in blue).
>
> ---
>
> >**Q2.** Were the minority samples only chosen based on the minority score? Did the authors perform any verification on whether they are all actually minorities? There are some samples shown for different minority scores, but I was wondering if there was a more systematic verification process or not.
>
> The real minority samples used in our quantative evaluations are chosen based on existing low-density metrics such as AvgkNN. For instance, the results in Table 1 are evaluated with the most unique instances that yield the highest AvgkNN values. See Section C for more details. The validation of minority score as a low-density metric can be found in Section D.2; see details therein.
>
> ---
>
> **References**
>
> [Ref-1] Diffusion models beat gans on image synthesis, NeurIPS 2021.
>
> [Ref-2] Improved denoising diffusion probabilistic models, ICML 2021.
>
> [Ref-3] High-Resolution Image Synthesis with Latent Diffusion Models, CVPR 2022.
>
> [Ref-4] Generating High Fidelity Data from Low-density Regions using Diffusion Models, CVPR 2022.

---

> > ### Comment · Reviewer_F8nE · 2023-11-23
> >
> > The authors have addressed most of my concerns, therefore I increase my rating.

---

### Official Review · Reviewer_mz51 · 2023-11-09

**Soundness:** 2 fair
**Presentation:** 3 good
**Contribution:** 2 fair
**Rating:** 3
**Confidence:** 5

**Summary:**

The paper claims that the distribution that diffusion learns tends to lead to the underrepresentation of certain parts of latent space. They tried to show and provide guidance to produce the underrepresented classes.
They used CelebA 64×64, ImageNet 64x64, LSUN Bedrooms 256x256, and a proprietary Brain MR dataset to show the effectiveness of their approach.

**Strengths:**

The introduction of metrics and the guidance to improve minority representation is interesting. The text is easy to follow and read, and the qualitative and quantitative results are intuitive. I enjoyed reading through the paper, despite some  questions raised through the arguments.

**Weaknesses:**

The paper interchangeably uses the general term of diffusion models and assumes that the Tweedy formula is a general form of any diffusion model.

The author cited the paper, Sehwag, V., Hazirbas, C., Gordo, A., Ozgenel, F. and Canton, C., 2022. Generating high fidelity data from low-density regions using diffusion models. In Proceedings of the IEEE/CVF Conference on Computer Vision and Pattern Recognition (pp. 11492-11501).
as a motivation "The principle guarantees their generated samples to respect the (clean) data distribution, which naturally makes the sampler become majority-oriented, i.e., producing higher likelihood samples more frequently than lower likelihood ones (Sehwag et al., 2022),"
but in the cited paper, they claimed that uniform sampling is the problem, and changing the sampling process could introduce novel, or as authors phrased samples from "long-tail" part of the distribution:
"We observe that uniform sampling from diffusion models predominantly samples from high-density regions of the data manifold. Therefore, we modify the sampling process to guide it towards low-density regions while simultaneously maintaining the fidelity of synthetic data."

The authors mentioned this baseline but only used it in one of the datasets, even though this work at least claimed that they tackled the problem of long-tailed distribution generation. In addition, in that dataset, the results are in 64x64, whereas Sehwag et al. mentioned that they produced 256x256 images after upscaling from 64x64.

This paper seems also doing similar approach, but has not been investgated.
Qin, Y., Zheng, H., Yao, J., Zhou, M. and Zhang, Y., 2023. Class-Balancing Diffusion Models. In Proceedings of the IEEE/CVF Conference on Computer Vision and Pattern Recognition (pp. 18434-18443).

**Questions:**

In Fig. 2, how the labels of small, moderate, and high minority scores are assigned in real data?
In Proposition 1, could you please clarify what you mean by "Assume that a given noise-conditioned score network sθ(xt, t) have enough capacity?" and how does it affect the proposed method?
Why did not you evaluate CIFAR-LT which is seemingly designed for such a task? (Cao, K., Wei, C., Gaidon, A., Arechiga, N. and Ma, T., 2019. Learning imbalanced datasets with label-distribution-aware margin loss. Advances in neural information processing systems, 3)

**Details Of Ethics Concerns:**

Please consider the legal and privacy issues of showing and evaluation on real brain MR images.

---

> ### Author Response · Authors · 2023-11-18
> **Official Comment by Authors (1/3)**
>
> We would like to thank your insightful and detailed comments, as well as your useful suggestions, which helped us to improve the manuscript. Below we provide point-by-point responses.
>
> ---
>
> >**W1.** The paper interchangeably uses the general term of diffusion models and assumes that the Tweedy formula is a general form of any diffusion model.
>
> We agree that our alternating use of the two general terms (i.e., diffusion and score-based models) may confuse the readers. That said, it is true that the diffusion models and score-based approaches are equivalent to each other up to parameterization (VP vs VE SDE) as demonstrated by Song et al [Ref-1]. Furthermore, we would like to kindly remind the reviewer that Tweedie’s formula is actually applicable to both VP-SDE (e.g., DDPM) and VE-SDE (e.g., Score-SDE) [Ref-2: DPS].
>
> ---
>
> >**W2.** The author cited the paper, Sehwag, V., Hazirbas, C., Gordo, A., Ozgenel, F. and Canton, C., 2022. Generating high fidelity data from low-density regions using diffusion models. In Proceedings of the IEEE/CVF Conference on Computer Vision and Pattern Recognition (pp. 11492-11501). as a motivation "The principle guarantees their generated samples to respect the (clean) data distribution, which naturally makes the sampler become majority-oriented, i.e., producing higher likelihood samples more frequently than lower likelihood ones (Sehwag et al., 2022)," but in the cited paper, they claimed that uniform sampling is the problem, and changing the sampling process could introduce novel, or as authors phrased samples from "long-tail" part of the distribution: "We observe that uniform sampling from diffusion models predominantly samples from high-density regions of the data manifold. Therefore, we modify the sampling process to guide it towards low-density regions while simultaneously maintaining the fidelity of synthetic data."
>
> Both our paper and Sehwag’s paper agree that the uniform sampling generates sample that follows the data distribution, which has more samples from high-density region. The reason we said it is a problem is that this does not favor to sample from minority regions, which we tried to overcome to generate minority samples.
>
> ---
>
> >**W3-1.** The authors mentioned this baseline but only used it in one of the datasets, even though this work at least claimed that they tackled the problem of long-tailed distribution generation.
>
> The reason we compare Sehwag et al. (2022) only in ImageNet is that their method was originally designed in the context of class-conditional settings. Nonetheless, to address your concern, we have extended Sehwag et al. (2022) into unconditional settings and compared the performance with ours. See below the table for instance where we provide comparison on CelebA:
>
> | Method                           | cFID   | sFID   | Prec  | Rec   |
> | -------------------------------- | ------ | ------ | ----- | ----- |
> | BigGAN        | 80.41  | 16.51   | **0.97**  | 0.19  |
> | DDPM         | 75.05  | 16.73   | **0.97**   | 0.23  |
> | DDPM-MLs          | 51.66  | 13.04  | 0.94   | 0.31  |
> | Sehwag et al. (2022)   | 27.97  | 10.17  | 0.82  | **0.42**  |
> | Ours          | **26.98**  | **8.23**  | 0.89  | 0.34  |
>
> We see the outperforming aspect of our method still persists even with an inclusion of Sehwag et al. (2022), demonstrating the advantage of ours as a minority generator. We now incorporated these results in our revision (Table 1 in Section 4.2, highlighted in blue).
>
> ---
>
> >**W3-2.** In addition, in that dataset, the results are in 64x64, whereas Sehwag et al. mentioned that they produced 256x256 images after upscaling from 64x64.
>
> As per your suggestion, we now updated our ImageNet results with upscaling to 256 x 256. See below for our new results on ImageNet 256:
>
> | Method                           | cFID   | sFID   | Prec  | Rec   |
> | -------------------------------- | ------ | ------ | ----- | ----- |
> | ADM [Ref-1]                       | 17.41  | 12.85  | **0.87**  | 0.35  |
> | Sehwag et al. (2022)              | **13.43**  | 9.88  | 0.85 | 0.42  |
> | Ours                                  | 13.83  | **7.82**  | 0.85  | **0.45** |
>
> We see the same trend of our performance benefit as in the 64 x 64 case, demonstrating the robustness of our approach not limited to low-resolution cases. We now included these results in our revision (Table 1 in Section 4.2, highlighted in blue).

---

> > ### Comment · Reviewer_mz51 · 2023-11-23
> > **W1**
> >
> > "We agree that our alternating use of the two general terms (i.e., diffusion and score-based models) may confuse the readers. It is true that the diffusion models and score-based approaches are equivalent to each other up to parameterization (VP vs VE SDE) as demonstrated by Song et al [Ref-1]."
> >
> > It is a rephrasing of my argument: "The paper interchangeably uses the general term of diffusion models and assumes that the Tweedy formula is a general form of any diffusion model."
> >
> > If the authors claimed that it is generalizable to all diffusion models, you need have full evaluation against all SOTA models.

---

> > ### Comment · Reviewer_mz51 · 2023-11-23
> > **W2**
> >
> > I mentioned that it might be the case that the sampling process and the objective that is used for the backward process might be the issue to the generation of the novel samples, but the authors did not answer this part.
> > "they claimed that uniform sampling is the problem, and changing the sampling process could introduce novel, or as authors phrased samples from "long-tail" part of the distribution"
> >
> > W2 rebuttal answer:
> > "Both our paper and Sehwag’s paper agree that the uniform sampling generates sample that follows the data distribution, which has more samples from high-density region. The reason we said it is a problem is that this does not favor to sample from minority regions, which we tried to overcome to generate minority samples."

---

> ### Author Response · Authors · 2023-11-18
> **Official Comment by Authors (2/3)**
>
> >**W4.** This paper seems also doing similar approach, but has not been investgated. Qin, Y., Zheng, H., Yao, J., Zhou, M. and Zhang, Y., 2023. Class-Balancing Diffusion Models. In Proceedings of the IEEE/CVF Conference on Computer Vision and Pattern Recognition (pp. 18434-18443).
>
> We would like to first kindly remind the reviewer that while sharing similar aspects of encouraging minority representations, the method by Qin et al. (2023) is distinct from ours specifically in terms of the dataset setting where they require the use of minority class labels that are often expensive to obtain in practice. In that sense, we believe making a direct comparison with ours is not possible. Nonetheless, we have conducted new experiments to explore the gap between ours and the method by Qin et al. (2023). Specifically on a new dataset, CIFAR-10-LT, we compare our method with Qin et al. (2023) as well as with the standard DDPM sampler; see below the table for the results:
>
> | Method                           | cFID   | sFID   | Prec  | Rec   |
> | -------------------------------- | ------ | ------ | ----- | ----- |
> | DDPM                             | 75.71  | 44.26  | **0.95**  | 0.23  |
> | Qin et al. (2023)                | 64.93  | **41.41**  | 0.92  | **0.36**  |
> | Ours                              | **63.34**  | 42.53  | **0.95**  | 0.24  |
>
> Notice that our sampler yields better (or comparable) performance than Qin et al. (2023), demonstrating the benefit of our method in boosting representations of minority instances without the help of related annotations. We now incorporated these new results in our revision (Table 7 in Section E.2, highlighted in blue).
>
> ---
>
> >**Q1.** In Fig. 2, how the labels of small, moderate, and high minority scores are assigned in real data?
>
> For the illustrations in Figure 2, we first measured the minority scores of all real samples (e.g., 50,000 training samples of CIFAR-10) via Eq. (7). The samples were then sorted based on their evaluated minority scores. We subsequently put instances yielding the lowest (highest) values of minority score in the left (right) column of Figure 2, while putting mid-level instances in the central column.
>
> ---
>
> >**Q2.** In Proposition 1, could you please clarify what you mean by "Assume that a given noise-conditioned score network sθ(xt, t) have enough capacity?" and how does it affect the proposed method?
>
> As mentioned in the proof of Proposition 1 (in Section A.1), the assumption supports the achievability of the optimal score network, which guarantees Tweedie’s formula (implemented using the score network) to exactly yield the posterior mean. Nonetheless, as already demonstrated in prior works [Ref-3], the posterior mean implemented with existing score architectures (e.g., U-Net) is sufficiently accurate to use for various applications [Ref-2,3], which corroborates with our results that exhibit the effectiveness of minority score in detecting low-likelihood instances (see Figures 2, 9, and 10 in our manuscript for instance). As a side note, most of the score method have the same assumption of the sufficient capacity to learn the score function.
>
> ---
>
> >**Q3.** Why did not you evaluate CIFAR-LT which is seemingly designed for such a task? (Cao, K., Wei, C., Gaidon, A., Arechiga, N. and Ma, T., 2019. Learning imbalanced datasets with label-distribution-aware margin loss. Advances in neural information processing systems, 3)
>
> As per your great suggestion, we now conduct new experiments on CIFAR-10-LT to further investigate the performance of ours. See the table below where we exhibit the performance of ours compared to a new baseline, the method by Qin et al. (2023). To measure the closeness to minorities in the long-tailed benchmark, we employ the most unique instances as baseline real data for computing the metrics (i.e., the same way of evaluation as in our manuscript):
>
> | Method                           | cFID   | sFID   | Prec  | Rec   |
> | -------------------------------- | ------ | ------ | ----- | ----- |
> | DDPM                             | 75.71  | 44.26  | **0.95**  | 0.23  |
> | Qin et al. (2023)                | 64.93  | **41.41**  | 0.92  | **0.36**  |
> | Ours                              | **63.34**  | 42.53  | **0.95**  | 0.24  |
>
> We see that our sampler performs better (or comparable) than the considered baselines, demonstrating the robustness aspect of ours that persists even under a challenging long-tailed setting. We now incorporated these new results in our revision (Table 7 in Section E.2, highlighted in blue).

---

> ### Author Response · Authors · 2023-11-18
> **Official Comment by Authors (3/3)**
>
> >**Flag For Ethics Review:** Yes, Privacy, security and safety, Yes, Legal compliance (e.g., GDPR, copyright, terms of use), Yes, Responsible research practice (e.g., human subjects, data release)
>
> >**Details Of Ethics Concerns:** Please consider the legal and privacy issues of showing and evaluation on real brain MR images.
>
> We would like to assure the reviewer that as clarified in Ethic Statement in our manuscript, the brain MRI data used in our experiments has received IRB approval, which includes meticulous attention to informed consent procedures and robust data anonymization protocols. Furthermore, our MRI dataset is exclusively for in-house purposes not is accessible to the public. Hence, we are confident that there are no lingering concerns pertaining to privacy, security, safety, and related matters.
>
> ---
>
> **References**
>
> [Ref-1] Score-based generative modeling through stochastic differential equations, ICLR 2021.
>
> [Ref-2] Diffusion posterior sampling for general noisy inverse problems, ICLR 2023.
>
> [Ref-3] Noise2score: tweedie's approach to self-supervised image denoising without clean images, NeurIPS 2021.

---

> ### Author Response · Authors · 2023-11-21
> **Dear Reviewer mz51**
>
> As the deadline for the Reviewer-Author discussion phase is fast approaching (there is only a day left), we respectfully ask whether we have addressed your questions and concerns adequately.

---

> ### Author Response · Authors · 2023-11-23
> **[Reminder] Summarization of our rebuttal**
>
> Dear Reviewer mz51,
>
> We have tried our best to address the concerns that you have raised. Specifically,
>
> 1. We have **extended the comparison** against Sehwag et al. (2022) on other settings like CelebA and ImageNet-256. We also incorporated the baseline that the reviewer suggested (i.e., Qin et al. (2023)).
> 2. We included experiments on an **additional benchmark** that the reviewer pointed out (i.e., CIFAR-10-LT). The results highlight that the performance benefit of our approach persists even on challenging long-tailed data.
> 3. We have clarified that there are **no remaining ethical issues** in the use of our brain MRI dataset, since it has received **IRB approval** and is not accessible to the public. We highlighted that we already mentioned this point in Ethical Statement in our submitted manuscript.
>
> We would like to gently remind you that the **end of the discussion period is imminent**. We would appreciate it if you could let us know whether our comments addressed your concerns.
>
> Best regards, Authors.

---

> > ### Comment · Reviewer_mz51 · 2023-11-23
> >
> > We have extended the comparison against Sehwag et al. (2022) on other settings like CelebA and ImageNet-256. We also incorporated the baseline that the reviewer suggested (i.e., Qin et al. (2023)).
> > "Thank you, but the results are marginally improved, and it cannot be generalized with limited evaluations, and the results are marginally improved over the only comparable work."
> >
> > The theoretical concerns were not answered.

---

### Author Response · Authors · 2023-11-18
**General response to all reviewers**

We would like to thank the reviewers for their insightful and comprehensive evaluations. We provide the results on some of the major concerns that were raised by the reviewers below. Point-by-point responses were also included as a reply to each reviewer.

---

**1. Comparison with additional baselines**

For a more thorough evaluation of the proposed method, we now incorporate several new baselines and compare the performance of ours with the additional methods. Specifically we exhibit three approaches herein.

The first one is an unconditional version of Sehwag et al. (2022) [Ref-1], which we extend their original method (specifically tailored for class-conditional settings) for the use in our unconditional experiments. We exhibit comparison results on the CelebA dataset herein. The second one is Qin et al. (2023) [Ref-2], an approach that leverages minority class labels at the training time to better learn representations of minorities. The comparison with Qin et al. (2023) is made on CIFAR-10-LT, our new benchmark to explore the performance under long-tailed benchmark. Lastly, the third baseline is LDM [Ref-3], a latent-based diffusion framework extensively used in various applications. We consider LSUN-Bedrooms to compare LDM with ours. See the tables below for the results:

| Method                           | cFID   | sFID   | Prec  | Rec   |
| -------------------------------- | ------ | ------ | ----- | ----- |
| BigGAN                            | 80.41  | 16.51   | **0.97**  | 0.19  |
| DDPM                             | 75.05  | 16.73   | **0.97**   | 0.23  |
| DDPM-MLs                          | 51.66  | 13.04  | 0.94   | 0.31  |
| Sehwag et al. (2022)              | 27.97  | 10.17  | 0.82  | **0.42**  |
| Ours                              | **26.98**  | **8.23**  | 0.89  | 0.34  |

| Method                           | cFID   | sFID   | Prec  | Rec   |
| -------------------------------- | ------ | ------ | ----- | ----- |
| DDPM                             | 75.71  | 44.26  | **0.95**  | 0.23  |
| Qin et al. (2023)                | 64.93  | **41.41**  | 0.92  | **0.36**  |
| Ours                              | **63.34**  | 42.53  | **0.95**  | 0.24  |

| Method                           | cFID   | sFID   | Prec  | Rec   |
| -------------------------------- | ------ | ------ | ----- | ----- |
| ADM [Ref-1]                    | 62.83 | 7.49 | 0.89  | 0.16  |
| LDM [Ref-3]                     | 63.61  | 7.19  | **0.90** | 0.13  |
| StyleGAN                          | 56.45  | **5.96**  | 0.87  | 0.13  |
| Ours                                  | **41.52**  | 6.67  | 0.87  | **0.10**  |

---

**2. Investigation on long-tailed benchmarks**

To further investigate the effectiveness of our approach on long-tailed data, we newly incorporate a famous benchmark specifically manipulated to exhibit long-tailed characteristics: CIFAR-10-LT. See the table below where we compare the performance of ours and an additional baseline, Qin et al. (2023). See the 1st bullet point above for details on the new baseline method. To measure the closeness to minorities in the long-tailed benchmark, we employ the most unique instances as baseline real data for computing the metrics (i.e., the same way of evaluation as in our manuscript):

| Method                           | cFID   | sFID   | Prec  | Rec   |
| -------------------------------- | ------ | ------ | ----- | ----- |
| DDPM                             | 75.71  | 44.26  | **0.95**  | 0.23  |
| Qin et al. (2023)                | 64.93  | **41.41**  | 0.92  | **0.36**  |
| Ours                              | **63.34**  | 42.53  | **0.95**  | 0.24  |


---

**References**

[Ref-1] Generating High Fidelity Data From Low-Density Regions Using Diffusion Models, CVPR 2022.

[Ref-2] Class-Balancing Diffusion Models, CVPR 2023.

[Ref-3] High-Resolution Image Synthesis with Latent Diffusion Models, CVPR 2022.

---

### Meta-Review · Area_Chair_AkNz · 2023-12-10

**Metareview:**

The paper presents a method that uses diffusion models to generate data from low density regions of the space. It uses Tweedie’s formula for denoising in doing this. Three of four reviewers thought the paper was worth accepting to the conference. They thought the technical background and presentation of the idea was well done, but did express some concern with the chosen baselines. After reading the response, two reviewers had their concerns sufficiently addressed.

**Justification For Why Not Higher Score:**

No reviewer expressed enthusiastic support via their score.

**Justification For Why Not Lower Score:**

Three reviewers thought the paper could be accepted and their reviews subjectively appear to outweigh the one negative review.

---

### Decision · Program_Chairs · 2024-01-16

Accept (poster)